# Omics-based Investigation of Diet-induced Obesity Synergized with HBx, Src, and p53 Mutation Accelerating Hepatocarcinogenesis in Zebrafish Model

**DOI:** 10.3390/cancers11121899

**Published:** 2019-11-28

**Authors:** Wan-Yu Yang, Pei-Shu Rao, Yong-Chun Luo, Hua-Kuo Lin, Sing-Han Huang, Jinn-Moon Yang, Chiou-Hwa Yuh

**Affiliations:** 1Institute of Molecular and Genomic Medicine, National Health Research Institutes, Zhunan 35053, Miaoli, Taiwan; cs081011@nhri.edu.tw (W.-Y.Y.); peishurao@gapp.nthu.edu.tw (P.-S.R.); hklin66@nhri.edu.tw (H.-K.L.); 2Department of Life Science, National Tsing-Hua University, Hsinchu 30070, Taiwan; 3Institute of Bioinformatics and Systems Biology, National Chiao Tung University, Hsinchu 30010, Taiwan; jimmy22452736@yahoo.com.tw (Y.-C.L.); gb921.tw@gmail.com (S.-H.H.); 4Department of Biological Science and Technology, National Chiao Tung University, Hsinchu 30010, Taiwan; 5Center for Intelligent Drug Systems and Smart Bio-devices, National Chiao Tung University, Hsinchu 30010, Taiwan; 6Institute of Bioinformatics and Structural Biology, National Tsing-Hua University, Hsinchu 30070, Taiwan; 7Program in Environmental and Occupational Medicine, Kaohsiung Medical University, Kaohsiung 80708, Taiwan

**Keywords:** obesity, nonalcoholic steatohepatitis (NASH), hepatocellular carcinoma (HCC)

## Abstract

The primary type of liver cancer, hepatocellular carcinoma (HCC), has been associated with nonalcoholic steatohepatitis, diabetes, and obesity. Previous studies have identified some genetic risk factors, such as hepatitis B virus X antigens, overexpression of SRC oncogene, and mutation of the p53 tumor suppressor gene; however, the synergism between diet and genetic risk factors is still unclear. To investigate the synergism between diet and genetic risk factors in hepatocarcinogenesis, we used zebrafish with four genetic backgrounds and overfeeding or high-fat-diet-induced obesity with an omics-based expression of genes and histopathological changes. The results show that overfeeding and high-fat diet can induce obesity and nonalcoholic steatohepatitis in wild-type fish. In HBx, Src (p53-) triple transgenic zebrafish, diet-induced obesity accelerated HCC formation at five months of age and increased the cancer incidence threefold. We developed a global omics data analysis method to investigate genes, pathways, and biological systems based on microarray and next-generation sequencing (NGS, RNA-seq) omics data of zebrafish with four diet and genetic risk factors. The results show that two Kyoto Encyclopedia of Genes and Genomes (KEGG) systems, metabolism and genetic information processing, as well as the pathways of fatty acid metabolism, steroid biosynthesis, and ribosome biogenesis, are activated during hepatocarcinogenesis. This study provides a systematic view of the synergism between genetic and diet factors in the dynamic liver cancer formation process, and indicate that overfeeding or a high-fat diet and the risk genes have a synergistic effect in causing liver cancer by affecting fatty acid metabolism and ribosome biogenesis.

## 1. Introduction

Hepatocellular carcinoma (HCC) is the fifth most common cancer and the third leading cause of mortality worldwide [1,2], and there is still no effective therapy available due to its heterogeneity [3]. Major risk factors for HCC include hepatitis B and hepatitis C virus infection, and aflatoxin contamination, as well as chronic alcohol consumption. HCC may be formed through hepatitis, fatty liver, and liver fibrosis, and eventually, develop into liver cancer. Hepatitis B (HBV) infection is a major risk factor of HCC [4], and X antigen (HBx) has been reported to be the most obvious carcinogen-induced liver cancer in mice [5] and zebrafish [6]. In human HCC patients, 75% of HCC cancer tissue were HBx positive [7], and 65.38% of HCC tissue were SRC positive in the Chinese population [8]. In the HBx-induced HCC mouse model, SRC was identified as a common regulator [9]. AFB1-induced p53 mutation at R249S mutation was highly associated with HCC [10]. HBx and TP53 R249S mutation were found in 77% of HCC patients in the West African population [11]. We demonstrated that HBx and SRC overexpression induced hepatocarcinogenesis in p53 mutant zebrafish [6]. Therefore, we generated a transgenic zebrafish model to reflect the genetic signature in human HCC patients.

HCC also has been associated with nonalcoholic fatty liver disease (NAFLD), nonalcoholic steatohepatitis (NASH), diabetes, and obesity [12,13,14]. Obesity is closely related to diabetes, chronic liver disease, and many cancers [15]. More importantly, obesity has also been identified as one of the main factors contributing to HCC [16,17,18]. According to a large epidemiological study, the number of obese children and adolescents between the ages of 5 and 19 increased tenfold in the past 40 years [19]. In Asia, obesity and diabetes are also increasing [20]. Metabolic risk factors such as fatty liver, high triglyceride levels, and diabetes mellitus are significantly associated with nonviral HCC in Taiwan [21]. With global anti-HBV vaccine and anti-HCV drugs, viral hepatitis-related liver cancer will gradually subside, and NAFLD-related HCC will become an important issue. NAFLD and NASH are metabolic diseases which are major drivers of HCC, and diet-induced obesity and high-fat diet is the cause of metabolic disorders. Genetic variants associated with obesity can be modified by obesogenic environments [22]. However, the synergistic effects between diet-induced obesity and genetic risk factors for liver disease and liver cancer are unclear. It is essential to understand the synergism between obesity and genetic risk factors and to develop therapeutic techniques derived from those discoveries.

Zebrafish is a vertebrate model with high relevance to humans; approximately 70% of human genes have at least one obvious zebrafish orthologue [23]. The well-developed gene transfer technology has boosted zebrafish as a prevalent research model in different research fields including human diseases, cancer studies, and drug screening [24,25,26]. Zebrafish are a model in vivo organisms for cancer research and drug identification, validation, and screening [27]. Zebrafish cancer models could be part of preclinical precision medicine approaches [28]. Even in the Cancer Moonshot project, zebrafish play an important role in developing new cancer technologies [29]. Previously, we developed some transgenic zebrafish to study hepatocarcinogenesis [6,30,31,32,33]. Due to the functional conservation in lipid metabolism, lipid biology, and glucose homeostasis, zebrafish have become a model for obesity and diabetes [34], and a great model for genetic- and diet-induced adiposity [35]. Overfeeding with 12 times the amount of *Artemia* for 8 weeks induced obesity and increased pathophysiological pathways in wild-type zebrafish, similar to mammalian obesity [36]. However, most of these studies lacked a global omics-based approach for a comprehensive analysis of obesity/NASH to HCC in the zebrafish model.

To investigate the synergistic mechanism of obesity/NASH/HCC in zebrafish models with three diet and four genetic risk factors, we developed an integrated omics computational model to examine their related genes, pathways, subsystems, and systems. We selected 30 samples for omics analysis using microarray (14 samples) and next-generation sequencing (NGS) (16 samples) to study the dynamic changes of pathways from normal diet (NOR), overfeeding (DIO) using diet-induced obesity, and high-fat-diet (FAT) treatment in four fish types: wild-type (WT), overexpression of hepatitis B virus X antigen (HBx) with p53 mutation (HBx(p53-)), overexpression of Src with p53 mutation (Src(p53-)), and overexpression of both HBx and Src with p53 mutation (HBx,Src(p53-)). Our results indicate that HBx, Src, and p53 mutations have a major impact on fat synthesis and carcinogenesis, and DIO or FAT synergizes with those risk factors in hepatocarcinogenesis. We believe that the observed mechanisms of interactions and the establishment of accelerated liver disease zebrafish models for drug screening can be of benefit to people.

## 2. Results

### 2.1. Overview of Omics-Based Investigation of Diet-Induced Obesity and Hepatocarcinogenesis in Zebrafish Model

We investigated the synergism between diet and genetic risk factors in hepatocarcinogenesis, and the main steps of our strategy are described in Figure 1A. First, three types of diet (NOR, DIO, and FAT) were given to zebrafish with four genetic backgrounds (WT, HBx(p53-), Src(p53-), and HBx,Src(p53-)) (Figure 1B). When compared to WT with a normal diet, DIO or FAT increased the expression of lipogenic factors and lipogenic enzymes, including 1-acylglycerol-3-phosphate acyltransferase (*agpat*), fatty acid synthase (*fasn*), and phosphatidate phosphatase (*pap*) using qPCR (Figure 1C) and hematoxylin and eosin (H&E) (Figure 1D). Comparing DIO and FAT to a normal diet, our global omics data analysis method identified differentially expressed genes (DEGs, such as *scd*, *gck*, and *slc40a1*; Figure 1E), significant pathways with *p*-value < 0.05 (e.g., steroid biosynthesis, FoxO signaling pathway, and insulin signaling pathway; Figure 1F). Finally, we found the metabolism and genetic information processing have an increasing tendency with the increase of risk factors in microarray (Figure 1G). These results show that our method can be useful for investigating gene-, pathway-, and system-level activation or inactivation of diet-induced obesity and hepatocarcinogenesis in zebrafish models.

### 2.2. Weight Changes in Zebrafish with Different Genetic Backgrounds by Three Feeding Methods

We first checked whether the weight of the fish increased with DIO and FAT for two months, starting at three months of age. For the wild-type, HBx(p53-), and Src(p53-) fish, the average weight from FAT was more significant compared with normal diet and DIO. In all fish species, the average weight gain from FAT was more significant than DIO. With a normal diet, the weight of the wild-type fish was the highest; in the DIO group, the weight of the Src(p53-) fish was the highest; in the FAT group, the weight of HBx(p53-) fish was the highest (Appendix A). We also noticed gender differences in the weight changes; weight gain was more significant in female than male fish (Appendix A).

### 2.3. Impact of Genetic Factors on Steatosis and Cell Proliferation

In HBx(p53-) and Src(p53-) normal feeding (NOR) fish for two months (equivalent to five-month-old fish), the expression of lipogenic enzyme or factors was higher than that of wild-type, and there was a significant or extremely significant difference (Appendix A). Those data indicate that HBx(p53-) and Src(p53-) are genetic risk factors for hepatocarcinogenesis, and they alone can slightly increase the expression of genes involved in fat synthesis in excessive or high-fat diets. Expression of HBx(p53-) or Src(p53-) has a major impact on the expression of lipogenic enzymes and factors. The effect of Src(p53-) is more dramatic than that of HBx(p53-). In HBx,Src(p53-) triple transgenic fish, there was no significant increased expression of lipogenic factors and enzymes, but the cell cycle/proliferation markers were significantly increased (Appendix A). Our results suggest that the impact of genetic factors on steatosis was more significant in the overexpression of HBx and Src alone in p53 mutant background. With the combination of HBx and Src in p53 mutant, the hepatocyte seems forwarded from overexpressing lipogenic enzyme/factors to cell cycle/proliferation, and we suspect the overexpression of oncogene-induced carcinogenesis.

### 2.4. Validation of Lipogenic Enzymes/Factors and Cell Cycle–Related Genes in Wild-Type Fish after Overfeeding

To examine the effects of diet-induced obesity on hepatocarcinogenesis, the expression patterns of lipogenic enzymes (*fasn*, *agpat*, and *pap*) and lipogenic factors (*pparg*, *srebf1*, and *chrebp*) were examined. We were also interested in examining the expression of the cell cycle/proliferation markers *ccne1*, *cdk1*, and *cdk2*.

In wild-type fish after eight weeks of feeding, for the lipogenic enzyme, DIO did not cause differences in the expression of *agpat* and *pap*, but decreased the expression of *fasn*. FAT increased the expression of *agpat* and *fasn*, but there was no significant difference in *pap* (Figure 2A). For the lipogenic factor, *pparg* and *srebf1* were highly expressed by DIO, and the expression of *srebf1* was more significantly increased in FAT than DIO. The expression of *chrebp* was decreased in both FAT and DIO after eight weeks (Figure 2B). The enhancement of lipogenic enzymes and factors was more obvious in female than male fish (Appendix A). In terms of cell cycle/proliferation markers, only *cdk1* was highly expressed; FAT did not show significant differences (Figure 2C). The increase of cell cycle/proliferation markers was more obvious in female than male fish (Appendix A).

### 2.5. Expression of Lipogenesis Factor, Lipogenesis Enzymes, and Cell Cycle–Related Genes in HBx(p53-), Src(p53-), and HBx,Src(p53-) Transgenic Fish after Overfeeding

In HBx(p53-) transgenic fish, DIO or FAT increased the expression of lipogenic enzymes and lipogenic factors (Figure 2D,E), and the enhancement was more obvious in females than in male fish (Appendix A). After DIO and FAT, only *cdk2* increased after eight weeks of high-fat diet, and *ccne1* and *cdk1* were significantly decreased after FAT diet (Figure 2F), and this enhancement seemed to be only in male fish (Appendix A).

In Src(p53-) transgenic fish, DIO or FAT decreased the expression of lipogenic enzymes (Figure 2G) and only FAT increased lipogenic factors (Figure 3H), and this enhancement seemed to be only in female fish (Appendix A). DIO and FAT did not increase the expression of cell cycle-related genes (Figure 3I) in either female or male fish (Appendix A). DIO and FAT in both HBx(p53-) and Src(p53-) transgenic fish had no further impact on the expression of cell cycle–related genes, indicating that the genetic and diet factors reached a plateau and may represent steatosis or early carcinogenesis.

From previous experiments, we found DIO and FAT diets exhibited similar effects on zebrafish, so for the triple transgenic fish, we only applied the DIO diet and compared with the normal diet. In HBx,Src(p53-) triple transgenic fish, DIO did not increase the expression of lipogenic enzymes and factors (Figure 2J,K) except for *srebf1*, and there was no gender difference (Appendix A). However, DIO further increased cell cycle–related gene expression (Figure 2L), and the enhancement was more obvious in female than male fish (Appendix A). Our data suggest that more genetic factors changed the genetic regulatory networks and, with diet-induced obesity, further promoted carcinogenesis.

### 2.6. Liver Pathology after Diet-Induced Obesity

We then examined liver specimens using H&E staining to verify the histopathological changes following different feeding treatments in transgenic fish. Representative images of H&E staining are shown in Figure 3A–C for WT, Figure 3E–G for HBx(p53-), Figure 3I–K for Src(p53-), and Figure 3M–O for HBx,Src(p53-), and the statistical analysis is shown in Figure 3D,H,L,P. We also analyzed the histopathological changes from overfeed for 16 weeks starting from 3 months of age and scarified the fish at 7 months (Figure 4). In WT fish, FAT increased steatosis, and both DIO and FAT caused slight hyperplasia (one fish out of 20 fish developed hyperplasia), which was more significant with prolonged feeding (Figure 4D). In HBx(p53-) and Src(p53-) transgenic fish, normal diet caused hyperplasia and FAT enhanced steatosis (Figure 3H,L). In HBx, Src(p53-) triple transgenic zebrafish, DIO accelerated HCC formation at five months of age and tripled the chances of getting HCC (Figure 3P). These histopathological features were consistent with the expression data from qPCR (Figure 2).

### 2.7. Global Omics Data Analysis

We proposed a systemic approach to analyze the whole-genome expression profile (i.e., microarray and NGS) of four types of fish after eight weeks of DIO and FAT (Appendix A). We first identified DEGs based on fold change >2, which included upregulated and downregulated genes in the four types of fish between normal and obesity diet (DIO and FAT). In microarrays of WT, HBx(p53-), Src(p53-), and HBx,Src(p53-) fish there were 767, 1032, 1557, and 1851 DEGs, respectively, and in NGS of WT, HBx(p53-), Src(p53-), and HBx,Src(p53-) fish there were 527, 500, 1352, and 1498 DEGs, respectively. The upregulated and downregulated genes underwent separate KEGG pathway enrichment analysis using hypergeometric distribution, and pathways with *p*-value < 0.05 were considered significant. Based on these pathway enrichments, we computed six KEGG system enrichments by using meta-z-scores.

At the system level, the meta-z-scores of microarray and NGS of the four types of fish are similar and consistently increase from WT, HBx(p53-), and Src(p53-) to HBx,Src(p53-), especially metabolism and genetic information processing (Figure 5A). In these two KEGG systems, the meta-z-scores of Src(p53-) and HBx,Src(p53-) are consistently higher than those of WT and HBx(p53-). These results imply that the pathways and genes of metabolism and genetic information processing have significant effects on hepatocarcinogenesis. Interestingly, the meta-z-score of Src(p53-) (11.48) is higher than that of HBx,Src(p53-) (9.38) in genetic information processing of NGS data. Based on Euclidean clustering, the omics data of the eight conditions (four genetic factors each for microarray and NGS) were divided into two cluster groups (Figure 5B). The WT and HBx(p53-) omics data of microarray (MIC) and NGS were clustered together, and the MIC and NGS omics data of Src(p53-) and HBx, Src(p53-) fish were clustered together. The results demonstrate that MIC and NGS have consistent biological mechanisms and the clustering result is in agreement with the histopathologic changes of the four genetic models.

To understand which genes and pathways in a system contribute to synergistic mechanisms of hepatocarcinogenesis, we analyzed pathway enrichment (*p*-value) of DEGs derived from eight kinds of omics data (four genetic factors of both MIC and NGS). We then selected 26 significant pathways (*p*-value < 0.05) that were consistent in both MIC and NGS (Figure 5C). According to the z-score profiles of these 26 pathways of four genetic factors, we observed several interesting results. First, some pathways (e.g., metabolic pathways, and amino sugar and nucleotide sugar metabolism) consistently increased from WT, HBx(p53-), and Src(p53-) to HBx,Src(p53-) fish. These pathways and DEGs play key roles in the normal state, steatosis, hyperplasia, and dysplasia toward HCC, and this result corresponds with H&E stain images and histopathologic change statistics (Figure 3D,H,L,P). The pathways involving central principles (i.e., protein processing in the endoplasmic reticulum, ribosome biogenesis in eukaryotes, and RNA polymerase) have high meta-z-scores in the genetic information processing system. Dysregulation of mRNA translation can be considered a cancer hallmark leading to aberrant proliferation. In the pathway of ribosome biogenesis, we found most of the genes (e.g., *nop56*, *dkc1*, and *emg1*) were downregulated and nonsignificant in WT and HBx(p53-), but upregulated in HBx, Src(p53-) (Appendix A).

Second, some pathways (e.g., pyrimidine metabolism, fatty acid metabolism, and steroid biosynthesis) consistently decreased in WT and HBx(p53-) fish and increased in Src(p53-) and HBx,Src(p53-) fish. Overexpression of HBx and mutation of p53 in WT fish led to slight hyperplasia, which caused the proliferation of normal cells and increased the cell numbers in the liver, but these pathways were inactivated in normal or slight hyperplasia state. Conversely, they activated serious hyperplasia and HCC state. The pyrimidine metabolism pathway contributed to form cancer mechanisms [37,38,39].

Furthermore, several pathways (e.g., glycolysis/gluconeogenesis, insulin signaling pathway, and insulin resistance) increased in WT and HBx(p53-) fish and decreased in Src(p53-) and HBx, Src(p53-) fish. Diabetes-related pathways were highly activated in WT fish with obesity diet, but inactivated in HBx, Src(p53-) fish. Many previous reports demonstrated that insulin resistance is a risk factor for cancer because it is a major component of metabolic syndrome, but the effect might decrease with the formation of HCC compared to WT with obesity diet [40]. The involved DEGs in insulin resistance pathway are listed in Appendix A.

Finally, some pathways (e.g., toll-like receptor (TLR) and p53 signaling pathways) consistently decreased in WT, HBx(p53-), Src(p53-), and HBx,Src(p53-) fish. In general, the p53 mutation in HBx(p53-), Src(p53-), and HBx, Src(p53-) fish inactivated the p53 signaling pathway. Interestingly, the trend of TLR inflammation was decreased in hepatocarcinogenesis in the zebrafish model. These results show that global omics data analysis can reveal synergistic mechanisms of hepatocarcinogenesis based on our method and omics data of zebrafish models with four genetic backgrounds.

### 2.8. Similarity Estimation among Genes, Pathways, and System Levels of Global Omics Data Analysis

To achieve a quantifiable outcome of similarity between microarray and NGS data in the four types of fish, we used the Jaccard coefficient to estimate the similarities of gene, pathway, and system levels. The Jaccard similarity index is defined as (1):(1)$$\mathrm{Jaccard}=\frac{S_{i}\cap S_{j}}{S_{i}\cup S_{j}}$$
where *S_i_* and *S_j_* are gene sets of the DEGs, significant pathways, or significant systems selected from microarray *i* and NGS *j* in a type of fish; *S_i_* ∩ *S_j_* and *S_i_* ∪ *S_j_* denote the intersection and union of DEGs, pathways, or systems between *S_i_* and *S_j_*, respectively. The Jaccard index of system level has the highest median (0.58); conversely, the Jaccard index of gene level has the lowest median (0.14) (Appendix A). This result implies that the genes selected from the microarray and NGS may be different, but they could participate in the same pathways or systems. Our method can eliminate the obstacles of different platforms (e.g., microarray and NGS) in omics data to identify the biological meanings reflecting hepatocarcinogenesis using different genetic risk factors and diets.

### 2.9. Identifying Potential Genes by Global Omics Data Analysis

To identify potential genes contributing to obesity, NASH, and HCC, we further developed maximum combined score (MCS) and average root combined score (ARCS) based on system-, subsystem-, pathway-, and gene- level global omics data analysis. We then evaluated the performance of these two scoring methods by 1960 (obesity) and 3592 (HCC) genes with gene–disease association score >0 as gold positive sets from the DisGeNET database [41]. Because the positive genes of DisGeNET are human, finally 828 (obesity) and 1400 (HCC) human genes were mapped into the zebrafish genes according to orthologues recorded in the KEGG database. Furthermore, we utilized precision to evaluate the performance of the three scoring systems, MCS, ARCS, and fold change. The results show that MCS and ARCS performed better than fold change for obesity and HCC (Figure 5). MCS and ARCS achieved similar performance for obesity (Figure 6A), and MCS showed the best performance among these methods for HCC (Figure 6B). Among these scoring methods, fold change was the worst for obesity and HCC genes. We show the top ranked 20 genes of MCS and the corresponding ranks of ARCS and FC in obesity and HCC, respectively. Among these 20 genes, 8 genes (e.g., *gck*, *scd*, and *pik3ca*) are related to obesity recorded in DisGeNET (Appendix A). Conversely, 7 and 5 genes are recorded in DisGeNET for the top ranked 20 genes from ARCS and FC, respectively. For the top ranked 20 genes of MCS, 11 genes (e.g., *sqlea*, *hmgcra*, and *mvda*) are related to HCC (Appendix A). In contrast to the MCS, 7 of the top ranked 20 genes from both ARCS and FC are recorded in DisGeNET.

There were 14 genes that were top scoring from microarray. These genes might be potential drug targets for the treatment of NASH, NAFLD, obesity and HCC. We further validate the mRNA expression levels using qPCR. We compared microarray data (Figure 7A–E) with qPCR (Figure 7F–J) on 5 of the top 14 genes from WT and different genetic background fish fed normal and obesity diets (Figure 7A–E).

We found that the mRNA expression levels of stearoyl-CoA desaturase (*scd*), which converts palmitic acid to monounsaturated fatty acids (Figure 7L), was increased not only in WT after overfeeding or high-fat diet, but also in HBx(p53-)-NOR and FAT and Src(p53-)-NOR and FAT fish (Figure 7F). Hepatic SCD has been associated with fatty liver, obesity, and metabolic diseases [42]; here we found mRNA level of *scd* was upregulated by over feeding and high-fat diet in WT, HBx(p53-) and Src(p53-) fish, which might indicates high hepatic *scd* plays a direct role in the development of fatty liver and development of metabolic disorders.

The mRNA expression levels of hepatic glucokinase (*gck*), which catalyzes the initial step of glucose utilization by liver, was increased in WT after overfeeding or high-fat diet, as well as in Src(p53-)-NOR and DIO fish (Figure 7G). This result is consistent with humans where the mRNA expression levels of GCK are associated with fatty liver and triglyceride contents [43]. Increased glucose uptake will facilitate formation of acetyl-CoA which then can be converted into lipid droplets by scd (Figure 7L).

The mRNA expression of acyl-Coenzyme a:cholesterol acyltransferase 2 (*acat2*), which converts acetyl-CoA to acetoacetyl-CoA and finally produce cholesterol, had increased expression in WT fed a high-fat diet, as well as in Src(p53-), Src(p53-)-DIO and HBx,Src(p53-) fish (Figure 7H). The direct role of ACAT2 linked to hepatic steatosis and glucose homeostasis was also proved in the mouse model [44].

The mRNA expression of phosphatidylinositol-4,5-bisphosphate 3-kinase catalytic subunit alpha (*pik3ca*), which is involved in PI3K signaling, was increased in WT fish fed a high-fat diet as well as overfed Src(p53-) fish (Figure 7I). PI3K/AKT signal increase de novo lipogenesis [45], and PIK3CA participated in the progression of NAFLD to NASH [46]. Activation of PI3K/Akt pathway is highly correlated with the development of liver fibrosis [47], this might explain why the upregulation of *pik3ca* was not in most of the fish with steatosis.

The expression of aldehyde dehydrogenase 7 family member A1 (*aldh7a1*), which participates in glucose, lipid, and amino acid metabolism, had increased expression in overfed and high-fat-diet-fed WT fish (Figure 7J). The upregulation of mRNA expression for the critical genes identified by microarray correlated to the steatosis during early hepatocarcinogenesis revealed by the histopathological diagnosis. Our global omics data analysis identified genes are connected to glucose and lipid metabolism also participating in NASH from the WT fish diet-induced obesity model (Figure 7L), and those genes might be potential drug targets for prevention of NASH, obesity and HCC. Our method discovered the potential genes in HBx(p53-) and Src(p53-) diet-induced obesity, which might represent the transition stages from NASH to early HCC (Appendix A), as well as the genes for HCC from overfed HBx,Src(p53) fish (Appendix A).

## 3. Discussion

Understanding the mechanisms of the transition from obesity/NASH to HCC is important for developing biomarkers and therapeutic strategies. We combined the phenotype data of one to three genetic risk factors with three diet types and cross-platform gene expression data to investigate the disease process and identify biomarkers or druggable genes using global omics data analysis and maximum combined score (MCS).

From our previous study, HBx(p53-) transgenic fish developed HCC at 11 months under a normal diet [6]. In this study, we fed the zebrafish for different diets at 3 months old for two months, and the sacrificed at the age of 5 months. We observed about 40% hyperplasia and no HCC, which is similar to what was found previously. Previously we found Src(p53-) transgenic fish developed HCC from 7 to 11 months, and only have hyperplasia at 5 months [6]. In this study, we also found Src(p53-) transgenic fish developed hyperplasia at 5 months of age, which is similar to previous study.

We have compared all HCCs from different genetic backgrounds and various diets to see how genetics influence HCC biology (Figure 4). Actually, the extensive diet treatment for 16 weeks for WT, HBx(p53-) and Src(p53-) promoted HCC formation; however, the HBx,Src(p53-) diet-induced obesity for 16 weeks reduced the HCC formation, which may due to the self-healing of zebrafish reported from various transgenic fish lines from our lab and others [6,32,33,48,49]. When we compared the HCC biology from different models, we saw the HBx, Src(p53-) diet-induced obesity for 8 weeks exhibited the most intensive HCC characteristics with more hepatocytes developed HCC. The HCCs developed from the extensive high-fat diet were also combined with steatosis.

We found that a high-fat diet and overfeeding can induce obesity and steatosis in wild-type fish by increasing the expression of genes involved in fat synthesis, and the change was more significant when there were no genetic risk factors. In the presence of two genetic risk factors, HBx(p53-) and Src(p53-), with a normal diet, hepatocytes underwent hyperplasia, and increased hyperplasia was only observed with prolonged feeding. Overfeeding- or high-fat-diet-induced overexpression of lipogenic factors and enzymes also increased steatosis measured by H&E stain. However, hyperplasia was already obvious with the normal diet, and the DIO diet can increase hyperplasia a little in HBx(p53-) fish.

In the HBx,Src(p53-) fish overfed for eight weeks, we observed that hyperplasia increased about threefold (from 21% in the normal diet to 62% in DIO), and HCC increased about threefold (from 7% in the normal diet to 23% in DIO). In HBx,Src(p53-) triple transgenic zebrafish, diet-induced obesity did not increase steatosis but accelerated HCC formation at five months of age, and the tripling enhanced the chances of getting HCC. In the global omics data analysis, we quantified the six KEGG systems to more comprehensively investigate the behaviors of the four types of fish between control and obesity groups regarding traditional pathways or genes. Our method initially offers direct clues for disease states or treatments. In addition, the pathway and gene-level showed details involving pathway activation/inactivation and gene upregulation/downregulation. Furthermore, if all three genetic risk factors were present HBx, Src(p53-) fish underwent earlier onset of liver cancer formation and threefold increased liver cancer incidence after eight weeks of overfeeding.

The progression from NAFLD to NASH in humans was originally proposed as a “two-hit hypothesis” [50], in which insulin resistance mediated increase of free fatty acids due to enhanced lipolysis was the first hit that leads to steatosis. The increased level of fatty acid oxidation enhancing oxidative stress was the second hit that triggers lipid peroxidation, inflammation, fibrosis, and carcinogenesis. Because the two-hit hypothesis is insufficient to explain the complicated mechanisms in NAFLD/NASH-HCC, the “multiparallel-hits hypothesis” was proposed [51], and has been recognized as the mechanism of NASH-HCC in humans [52], in which hepatic inflammation was the first cause, and numerous conditions (including genetic variations, abnormal lipid metabolism, oxidative and/or endoplasmic reticulum stress (ER stress), mitochondrial dysfunction, altered immune responses, and imbalance in gut microbiota) act in parallel.

In the molecular levels, complex changes in signaling pathways due to genetic and epigenetic changes mediate metabolism dysregulation and cell proliferation [53]. The pro-inflammatory cytokines IL-6 activates IAK/STAT3, phosphatidylinositol 3-kinases (PI3K)/AKT/mTOR, mitogen-activate protein kinase (MAPK) pathway, and TGF-*β* and Wnt/*β*-catenin that regulate proliferation and energy metabolism in the cell were reported.

The combination of endoplasmic reticulum stress and a high-fat diet (HFD) can lead to HCC through a number of underlying mechanisms. HFD can produce moderate ER stress, which increases lipogenesis and hepatic steatosis, while they increase reactive oxygen species (ROS) and oxidative stress and subsequently cause genomic instability, leading to the death of hepatic cells and the release of inflammatory factors that stimulate hepatocyte proliferation leading to HCC [52].

Using a mouse model fed with high-fat-non-cholesterol versus high-fat-high-cholesterol, scientists have found high-cholesterol promotes NASH development. Upregulation of the metabolic genes (*ALDH18A1*, *CAD*, *CHKA*, *POLD4*, *PSPH*, and *SQLE*) and aberrant expression of cancer-related genes (*ALCAM*, *ITGA6*, *DDIT3*, *MAP3K6*, and *PAK1*) were found in mouse fed with high-fat-high-cholesterol similar to human NASH-HCCs [54]. Another female mice model fed with Western diet (WD)-induced NASH increased the expression of genes, including steatosis (SFA, MUFA, MUFA-containing di- and triacylglycerol), inflammation (*TNFα*), oxidative stress (*Ncf2*), and fibrosis (*Col1A*) via lipidomics and transcriptomic approach [55]. Our results revealed that pathways of fatty acid metabolism and steroid biosynthesis are activated during hepatocarcinogenesis which is consistent with previous results. In our results, we also found glycolysis/gluconeogenesis, insulin signaling pathways, and insulin resistance pathways were increased in WT and HBx(p53-) DIO fish. Our results also indicated that ribosome biogenesis is activated during hepatocarcinogenesis, which has not been mentioned in previous studies. Moreover, we found that genes related to glucose and lipid metabolism are overexpressed in WT diet-induced obesity, revealing glucose uptake from overfeeding will link to lipogenesis. This finding is novel and might explain the molecular mechanisms for overfeeding causing NASH.

Based on 26 significantly consistent pathways (*p*-value < 0.05) in both MIC and NGS (Figure 5C), some of which are related to immune responses in four genetic background zebrafish, we summarize the observations as follows: First, toll-like receptor signaling pathway, playing key roles in the immune system, consistently decreased in WT, HBx(p53-), Src(p53-), and HBx, Src(p53-) fish. We found the gene expressions of *tlr5a* and *tlr5b* were up-regulated in WT, and gene *nfkbiaa* was up-regulated in both WT and HBx(p53-). These genes are highly related to the immune response. In addition, several pathways (e.g., NOD-like receptor signaling pathway and RIG-I-like receptor signaling pathway), which belong to the immune system based on the KEGG database, have similar trends in omics data. Second, insulin signaling and insulin resistance pathways were considered to participate in the regulation of islet endocrine influenced by the immune system [56,57]. They increased in WT and HBx(p53-) fish and decreased in Src(p53-) and HBx,Src(p53-) fish. Third, the pathway of proteasome consistently decreased in WT and HBx(p53-) fish and increased in Src(p53-) and HBx,Src(p53-) fish. It regulates the immune system by degrading immune and inflammatory mediators [58]. Interestingly, the trend of the proteasome is opposite to the trend of the toll-like receptor signaling pathway. Finally, the metabolic pathways (e.g., fatty acid degradation) consistently increased from WT, HBx(p53-), and Src(p53-) to HBx,Src(p53-) fish. Immune responses could be potentially modified by fatty acids, and the modifications include the organization of lipids in the cells and interaction with nuclear receptors [59]. These results imply that the immune system plays a key role in diet-induced obesity and accelerating hepatocarcinogenesis in zebrafish.

Our method has several limitations, challenges, and perspectives. First, MCS may prefer well-studied genes, since only genes recorded in the KEGG database were considered in this study. Second, the scores of the pathway (z-scores) and system (meta-z-scores) represent enrichment computed by significant up- and downregulated genes. Therefore, a high z-score may be caused by downregulated genes. For example, most DEGs in the steroid biosynthesis pathway in hepatocarcinogenesis (i.e., from WT to HBx,Src(p53-)) are downregulated (Appendix A).

## 4. Materials and Methods

### 4.1. Zebrafish Maintenance and Transgenic Zebrafish Lines

The AB zebrafish strain (Danio rerio), Tg(fabp10a:Src(p53-)), Tg(fabp10a:HBx(p53-)), and Tg(fabp10a:HBx,Src(p53-)) were used in this study. The AB strain was obtained from the Zebrafish International Resource Center (ZIRC, Eugene, OR, USA), and the other transgenic lines were bred in our laboratory. All fish were maintained in the Zebrafish Core Facility at the National Health Research Institute (NHRI) in Taiwan. All experiments involving zebrafish were approved by the Institution Animal Care and Use Committee (IACUC) of the NHRI (protocol No. NHRI-IACUC-106119-A). The embryos were cultured in the laboratory and raised at 28 °C as described previously [60].

Four types of fish, 3 months old, were treated with 3 feeding methods: normal diet, overfeeding, or high-fat diet. After 8 weeks of feeding, fish were weighed, and liver tissue from about 20 fish was collected. The weight changes of each group were measured and recorded weekly. One-third of the collected liver tissue was taken for RNA extraction, and the resulting mRNA was reverse transcribed into cDNA and subjected to qPCR and analyzed for expression of marker genes for lipogenic factors, lipogenic enzymes, and cell cycle/proliferation. One-third of the liver tissue was taken for paraffin embedding and sectioning, and H&E staining was performed. The stained sections were photographed and the pathological features of liver tissue were analyzed.

### 4.2. Feeding Method

Wild-type zebrafish overfed with 12 times the amount of *Artemia* than those fed with 5 mg dry weight/day/fish for 8 weeks became obese and had increased expression of pathophysiological pathways similar to mammalian obesity [36]. We followed the diet-induced-obesity method and fed wild-type, HBx(p53-), Src(p53-), and HBx, Src(p53-) fish with 12 times the amount of Artemia or a high-fat diet for 8 weeks. All groups were fed a spoonful of powdered feed (about 0.22 g) at 09:00 every day. Newly hatched brine shrimp were collected into a 50 ml centrifuge tube and placed in a dark room to settle to the bottom. The volume of the brine shrimp was concentrated to 10 mL, and the supernatant was aspirated for a total volume of 48.5 mL, shaken well before feeding. The normal feeding group (NOR) were fed 0.5 mL of *Artemia* once a day. The DIO groups received 2 mL of *Artemia* three times a day. The high-fat-feed group (FAT) were fed with 0.5 mL of *Artemia* once a day, and one spoonful of high-fat fish food (about 0.22 g) three times daily. The feeding schedule is listed in Table 1.

### 4.3. Body Weight Measurement and Liver Specimen Collection

Weekly body weights were measured for 8 weeks and recorded as continuous changes in body weight. To do this, we took the fish from the tank and anesthetized them with 1X anesthetic solution (10 mg Tricaine powder, 48 mL ddH2O, 2 mL Tris (1 M, pH 9.5), neutralized to pH 7.0–7.5), picked up all the fish and drained them with paper towels, then put them into a 200 mL beaker on an electronic scale and recorded the total weight. Then we put the fish back into the tank, filled with clean water, for them to recover and be returned to the original fish tank.

Endpoint body weights were measured after 8 weeks of feeding. We immersed the fish from which liver specimens would be collected into the anesthetic. After they were anesthetized, we drained the fish with paper towels, then put them into a 200 mL beaker on an electronic scale and recorded the total weight. After reading the numbers, we placed them in a Petri dish and started collecting the liver samples by cutting the fish belly with dissecting scissors, removing the liver, removing other internal organs, and dividing the liver into 3 equal parts. One-third of the liver was placed in a microcentrifuge tube for RNA extraction, and beads for homogenization (0.5 mm) were added. For histopathological analysis, one-third of the liver was placed in a microcentrifuge tube to which 10% formalin had been added, and then paraffin-embedded and sectioned by the NHRI pathological core laboratory. For protein analysis, one-third of the liver was placed in a microcentrifuge tube containing a protein extracting reagent and homogenized beads (0.5 mm). After all the fish were processed, the liver specimens were put in liquid nitrogen and stored at −80 °C.

### 4.4. RNA Analysis of Gene Expression

Liver tissue was ground in a homogenizer and RNA was extracted using the RNAspin Mini RNA Isolation Kit (GE Healthcare). Reverse transcription PCR (RT-PCR) was performed using a High-Capacity RNA-to-cDNA™ Kit (Life Technologies). Then 10 μL of 2× RT buffer and 1 μL of 20× enzyme mix were added to 2 μg of RNA sample, and nuclease-free H_2_O was added for a final volume of 20 μL. Samples were incubated in a polymerase chain reactor for the reverse transcription reaction under the following conditions: 37 °C for 60 min, 95 °C for 5 min, and finally back to 4 °C. The cDNA was subjected to quantitative PCR or stored at −80 °C.

### 4.5. Quantitative PCR (qPCR)

The resulting first-strand cDNA was used as a template for qPCR in triplicate using the KAPA SYBR^®^ FAST qPCR Kit Master Mix (2×) ROX Low (Kapa Biosystems, USA) with an ABI PRISM 7900 PCR System. Ten genes (actin, *pparg*, *srebf1*, *chrebp*, *fasn*, *pap*, *agpat*, *ccne1*, *cdk1*, *cdk2*) were detected per sample. The specific primers used in the qPCR are listed in Appendix A. All experiments were performed in triplicate, and mean values were obtained. Actin was used as internal control. The reaction parameters were set as follows:(1)50 °C for 2 min, 95 °C for 5 min, 4 °C thereafter(2)95 °C for 10 min(3)95 °C for 15 sec, 60 °C for 1 min (40 cycles)(4)95 °C for 15 sec, 60 °C for 15 sec, 95 °C for 15 sec

After normalization to actin as internal control, the expression ratio between experimental and control groups was calculated using the comparative Ct method. The relative expression ratio (fold change) was calculated based on ΔΔCt, which was (Ct_(target)_ − Ct_(actin)_) of the experimental group−(Ct_(target)_ − Ct_(actin)_) of the control group, where fold change = 1.94^−ΔΔCt^. The fold changes were calculated considering the qPCR reaction efficiency. At least 3 independent samples were used for the qPCR, and medians and standard errors were calculated and are presented as median ± standard error.

### 4.6. Histopathological Analysis

For histopathological analysis, the tissues were fixed in a 10% formalin solution (Sigma-Aldrich Inc., St. Louis, MO, USA), embedded in paraffin, sectioned at a thickness of 5 μm, mounted on Poly-L-lysine-coated slides, and stained with hematoxylin and eosin (H&E). After xylene and ethanol were used to gradually dewax and rehydrate, the hematoxylin and eosin dye was added, then they were gradually dehydrated and sealed. The pathological analysis was judged as follows. Normal: normal cells are arranged neatly, the size of the nucleus is similar, and the cytoplasmic ratio is not high. Steatosis: a vacuole composed of fatty oil droplets in the cytoplasm of hepatocytes can be seen. Hyperplasia: there are large and slightly abnormal nuclei with a high nucleus-to-cytoplasm ratio. Dysplasia: cells that have changed morphology, with larger nuclei and distinct nucleoli, are seen. HCC: there are large pleural and enlarged nuclei, in which nucleoli can clearly be observed.

### 4.7. Statistical Analysis

The body weight and qPCR data were analyzed using SPSS 17.0 (SPSS, Inc.) and Prims GraphPad. The statistical analysis was performed using a two-tailed Student’s t-test. In all statistical analyses, *p*-values < 0.05 were considered to be statistically significant and are presented as: * *p* ≤ 0.05; ** *p* ≤ 0.01; *** *p* ≤ 0.001; and **** *p* ≤ 0.0001.

### 4.8. GeneTitan™ Array and RNA-seq for Gene Expression Profiling

In total, 14 and 16 selected RNA samples from 8 weeks of feeding were subjected to GeneTitan microarray and RNA sequencing (RNA-seq) analysis, respectively. RNA-seq used next-generation sequencing (NGS) to analyze gene expression of samples. ZebGene 1.1 ST Array Plates (Affymetrix, USA) and Illumina HiSeq 4000 platform with 150 bp paired-end reads (Illumina, USA) were used for the whole-genome transcriptome analysis. The raw data of the microarray and RNA-seq have been submitted to the NCBI Gene Expression Omnibus (GEO) (http://www.ncbi.nlm.nih.gov/geo/) under accession code GSE134496 as SuperSeries. The SuperSeries is composed of two SubSeries as GSE134494 (microarray) and GSE134495 (RNA-seq).

### 4.9. Global Omics Data Analysis

For microarray, normalization of gene-level data was performed by the robust multi-array average (RMA) method [61] in R [23] using the oligo package [62]. For RNAseq of NGS, the reads were aligned with the zebrafish reference genome (GRCz10/danRer10) [63] using the HISAT package [64] and the mapped reads were assembled into transcripts using StringTie [65]. Next, the expression levels of all transcripts were estimated and calculated based on fragments per kilobase of transcript per million fragments mapped (FPKM) by StringTie and the Ballgown package in R, respectively. The average mapping rate of reads was 89% (Appendix A).

For microarray and NGS analysis, we first identified the differentially expressed genes (DEGs) with fold change ≥2 using the limma package [66]. Based on these DEGs, the hypergeometric distribution [67,68] was used for pathway enrichment analysis, and pathways with *p*-value < 0.05 were considered significant. Here, the *p*-value is calculated as (2):(2)$$p=\overset{n}{\underset{i=x}{\sum }}\frac{\left(\begin{array}{c} M\\ i \end{array}\right)\left(\begin{array}{c} N-M\\ n-i \end{array}\right)}{\left(\begin{array}{c} N\\ n \end{array}\right)}$$
where *N* is the number of genes recorded in KEGG, *M* is the number of DEGs, *n* is the number of genes in the specific KEGG pathway, and *i* is the number of genes that are DEGs in the specific KEGG pathway. *N* and *n* were obtained from the database for annotation, visualization, and integrated discovery (DAVID) v6.8 [69]. The *p*-value of each pathway underwent a z-score transformation using the standard normal distribution. We then computed the subsystem- and system-level meta-z-scores of 58 KEGG subsystems and 6 systems: metabolism, genetic information processing, environmental information processing, cellular processes, organismal systems, and human diseases. The z-score is defined as (3): (3)$$\mathrm{meta}-z=\frac{\sum_{i=1}^{n}z_{i}}{\sqrt{n}}$$
where *Z_i_* is the z-score of the pathway *i*, and *n* is the total number of pathways in a subsystem or system. The meta-z-score reflects the significance (enrichment) of a subsystem/system for a specific disease state, such as HBx(p53-), Src(p53-), or HBx,Src(p53-) fish with diet-induced obesity.

### 4.10. Scoring Function for Potential Gene Selection

To identify potential genes for obesity/NASH to HCC, we developed two scores: maximum combined score (MCS) and average root combined score (ARCS). For gene *i* involving *N* pathways, MCS and ARCS are computed as (4) and (5):(4)$$\mathrm{MCS}=max_{x,y∋j=1}^{N}\left\{Z_{sys_{x}}+Z_{sub_{y}}+Z_{path_{j}}\right\}+S_{N}+S_{FC}$$
(5)$$\mathrm{ARCS}=\frac{\sum_{x,y∋j=1}^{N}\left(Z_{sys_{x}}+Z_{sub_{y}}+Z_{path_{j}}\right)+N*\left(S_{N}+S_{FC}\right)}{\sqrt{N}}$$
where $Z_{sys_{x}}$ and $Z_{sub_{y}}$ are the meta-z-scores of system *x* and subsystem *y*, respectively; *S_N_* and *S_FC_* are the *N* pathways and the |*log_2_FC*| between control and obesity groups, respectively. Then, each scoring term is normalized ranging from 0 to 1 by min-max normalization. Therefore, MCS values range from 0 to 5, and the range of ARCS is dependent on *N*. $Z_{sys_{x}}$ and $Z_{sub_{y}}$ are defined as (6) and (7):(6)$$Z_{sys_{x∋j}}=\frac{\sum_{j=1}^{N_{x}}z_{j}}{\sqrt{N_{x}}}$$
(7)$$Z_{sub_{y∋j}}=\frac{\sum_{j=1}^{N_{y}}z_{j}}{\sqrt{N_{y}}}$$
where *z_j_* is the z-score of pathway *j* involved in system *x* and subsystem *y*; *N_x_* and *N_y_* are the numbers of pathways of gene *j* involved in system *x* and subsystem *y*. Based on the *Z_sys_* and *Z_sub_* values of 6 systems and 58 subsystems, *Zsys_x_* and *Zsub_y_* were normalized from 0 to 1 by min-max normalization. *S_N_* is given as (8):(8)$$S_{N}=\frac{N-min\left\{N_{G}\right\}}{max\left\{N_{G}\right\}-min\left\{N_{G}\right\}}$$
where $max\left\{N_{G}\right\}$ and $min\left\{N_{G}\right\}$ are the maximum and minimum involved pathways for 5023 genes recorded in the KEGG database. $S_{FC}$ is defined as (9):(9)$$S_{FC}=\frac{log_{2}FC-min\left\{log_{2}FC_{G}\right\}}{max\left\{log_{2}FC_{G}\right\}-min\left\{log_{2}FC_{G}\right\}}$$
where $max\left\{log_{2}FC_{G}\right\}$ and $min\left\{log_{2}FC_{G}\right\}$ are the maximum and minimum |*log_2_FC*| values for 5023 genes recorded in the KEGG database. The schematic diagram is shown in Appendix A.

Furthermore, we utilized precision to evaluate the performance of MCS, ARCS, and fold change to select potential genes using the positive sets selected from the DisGeNET database. Here, precision is defined as TP/(TP + FP), where TP and FP are the numbers of true positive and false-positive genes, respectively.

### 4.11. GeneTitan™ Array for Gene Expression Profiling

ZebGene 1.1 ST Array Plates (Affymetrix, USA) were used for whole-genome transcriptome analysis. Using Transcriptome Analysis Console (TAC) software, differentially expressed genes of DIO versus control were identified. Expression analysis settings were as follows: gene-level fold change <−2 or >2, gene-level *p*-value <0.05, ANOVA method: ebayes. The protein–protein interactions were analyzed using NetworkAnalyst (http://www.networkanalyst.ca/) and pathway activations was selected and matched according to the Kyoto Encyclopedia of Genes and Genomes (KEGG) database.

## 5. Conclusions

Against a normal genetic background, diet-induced obesity will increase the chance of fatty liver with a low incidence of hyperplasia. Two genetic risk factors together with diet-induced obesity will increase the chances of having fatty liver and hyperplasia. With three genetic risk factors, the probability of cancer formation is higher, and diet-induced obesity will accelerate cancer formation threefold. The genetic background from normal to two genetic risk factors with diet-induced obesity is a similar process of hepatocarcinogenesis from obesity/NASH, verified by histopathological analysis. The metabolic and genetic information processing systems are affected significantly as the genetic risk factors increase in cross-platform data using global omics data analysis. The insulin signaling pathway plays a key role in the normal genetic background with diet-induced obesity, but it seems to be not important in two genetic risk factors. Furthermore, we provide a maximum combined score to select potential genes (i.e., *scd*, *gck*, *acat2*, *pik3ca*, and *aldh7a1*) that participate in the obesity/NASH to HCC process, verified by qPCR. Our approach is useful for the development of biomarkers and therapeutic targets by considering multiple dimensions. Our zebrafish model and methods provide an explanation for the synergism between genetic risk factors and diet-induced obesity.

## Figures and Tables

**Figure 1 cancers-11-01899-f001:**
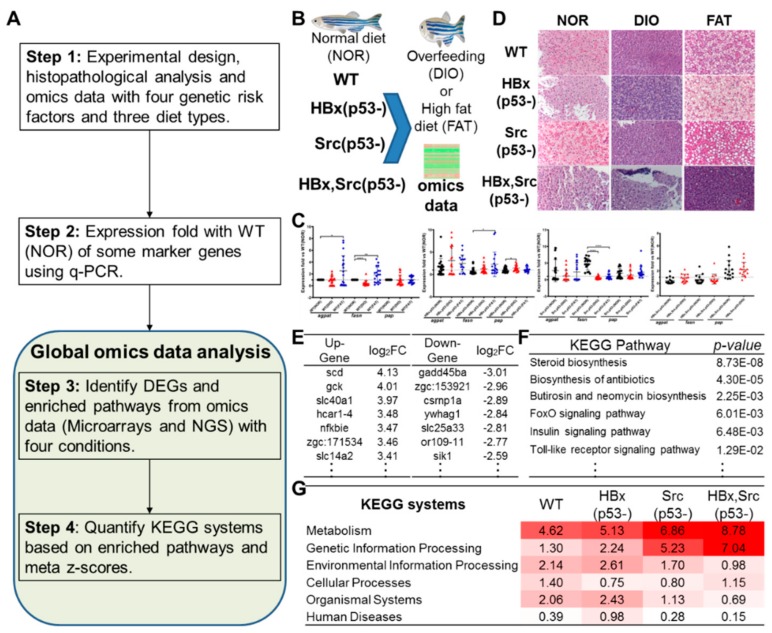
Overview of omics-based investigation of hepatocarcinogenesis in zebrafish models. (**A**) Main procedure. (**B**) Experimental design of omics data in zebrafish with four genetic backgrounds: WT, HBx(p53-), Src(p53-), and HBx,Src(p53-) with diet-induced obesity. (**C**) qPCR of some gene markers in the four models. (**D**) Hematoxylin and eosin (H&E) stain images of the four zebrafish models with normal diet (NOR), overfeeding (diet-induced obesity, DIO), and high-fat diet (FAT). (**E**) Selected differentially expressed genes (DEGs). (**F**) Some enriched KEGG pathways of selected DEGs. (**G**) Meta-z-scores of six KEGG systems of four zebrafish models.

**Figure 2 cancers-11-01899-f002:**
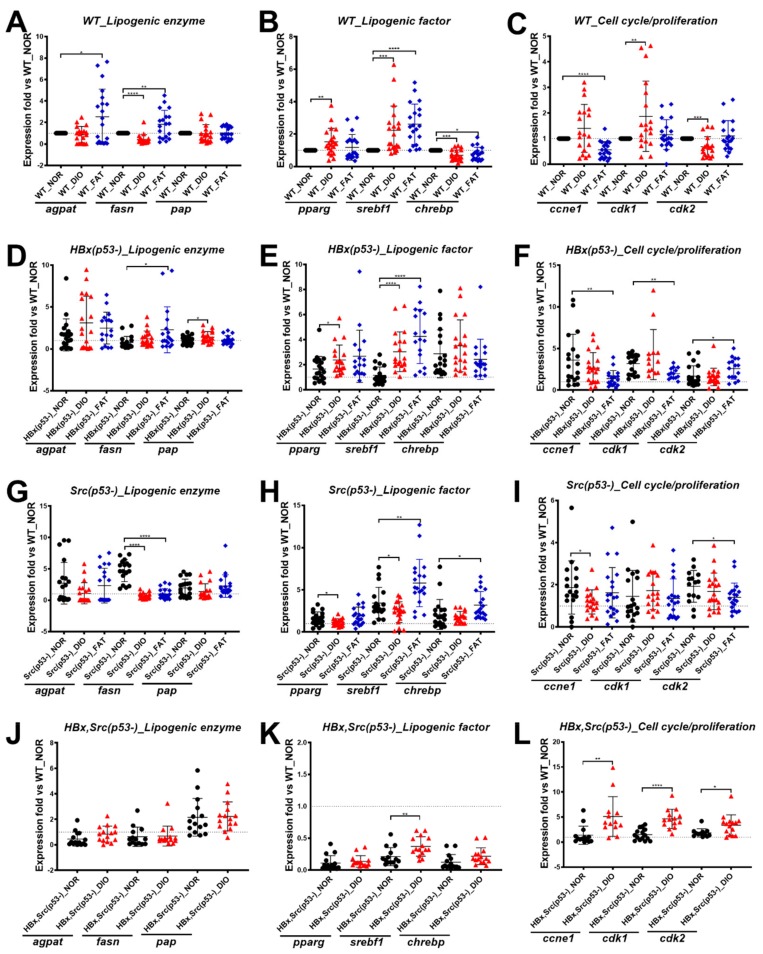
Expression of selected markers in various genetic background zebrafish fed with different diets. Expression of lipogenic enzymes (*agpat*, *fasn*, and *pap*), lipogenic factors (*pparg*, *srebf1*, and *chrebp*), and cell cycle/proliferation-related genes (*ccne1*, *cdk1*, and *cdk2*) in (**A**–**C**) WT, (**D**–**F**) HBx(p53-), (**G**–**I**) Src(p53-), and (**J**–**L**) HBx,Src(p53-) fish after eight weeks of normal diet (NOR), overfeeding (diet-induced obesity, DIO), or high-fat diet (FAT). The number of fish is 20 for each group, and the number of experimental replicates for qPCR analysis is three. Expression fold change compared to WT_NOR control. Statistical analysis of results was performed using a two-tailed Student’s t-test. Asterisks (*) represent level of significance: * *p*-value ≤ 0.05; ** *p*-value ≤ 0.01; *** *p*-value ≤ 0.001; **** *p*-value ≤ 0.0001.

**Figure 3 cancers-11-01899-f003:**
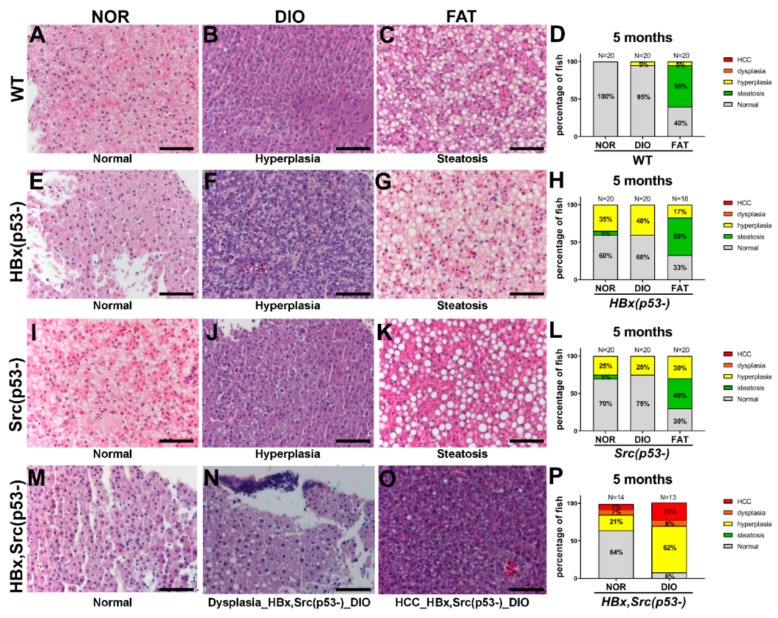
Histopathological changes in various genetic background zebrafish fed with different diets. Representative H&E stain images and histopathologic change statistics of (**A**–**D**) WT, (**E**–**H**) HBx(p53-), (**I**–**L**) Src(p53-), and (**M**–**P**) HBx, Src(p53-) fish after 8 weeks of normal diet (NOR), overfeeding (diet-induced obesity, DIO), or high-fat diet (FAT) starting at 3 months of age, and scarified at 5 months. The histopathologic change consists of percentages in five states: normal, steatosis, hyperplasia, dysplasia, and hepatocellular carcinoma (HCC). The number of fish is represented as N on top of each bar. The scale bar is 50 μm.

**Figure 4 cancers-11-01899-f004:**
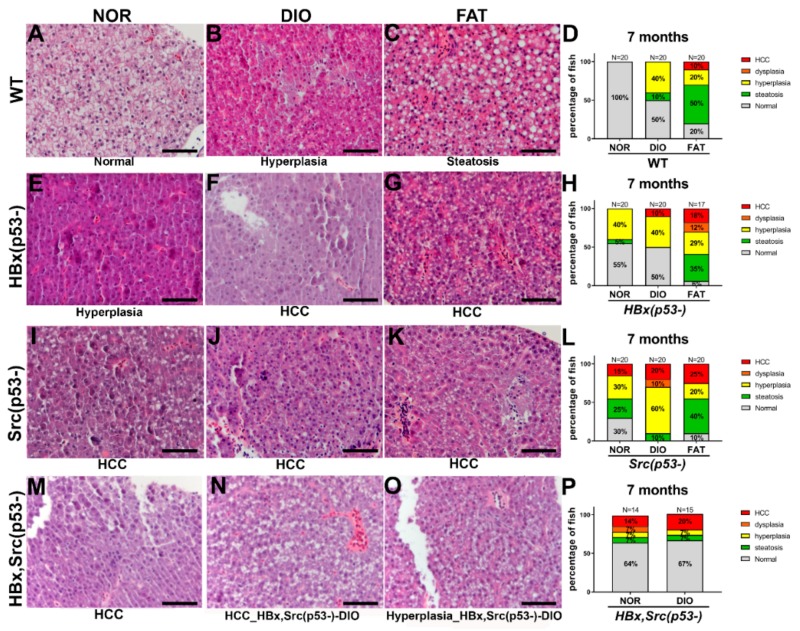
Histopathological changes in various genetic background zebrafish fed with different diets. Representative H&E stain images and histopathologic change statistics of (**A**–**D**) WT, (**E**–**H**) HBx(p53-), (**I**–**L**) Src(p53-), and (**M**–**P**) HBx,Src(p53-) fish after 16 weeks of normal diet (NOR), overfeeding (diet-induced obesity, DIO), or high-fat diet (FAT) starting at 3 months of age, and scarified at 7 months. The histopathologic change consists of percentages in five states: normal, steatosis, hyperplasia, dysplasia, and hepatocellular carcinoma (HCC). The number of fish is represented as N on top of each bar. The scale bar is 50 μm.

**Figure 5 cancers-11-01899-f005:**
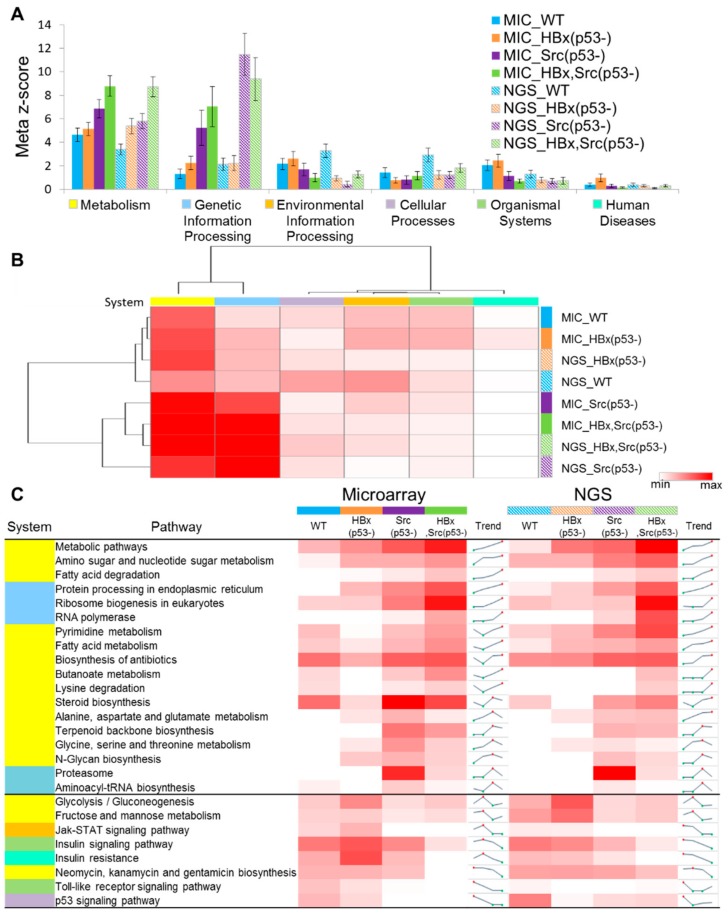
Statistics of system-level meta-z-scores and pathway z-scores for the four genetic fish models. (**A**) Meta-z-scores and (**B**) heatmap of six KEGG systems of the 30 samples with four models and three diet types in microarray and NGS. WT: blue; HBx(p53-): deep orange; Src(p53-): purple; and HBx,Src(p53-): green. MIC, microarray; NGS, next-generation sequencing. Red and white denote high and low meta-z-scores, respectively. (**C**) Pathway z-score distributions and trends of microarray and NGS in the four models. For KEGG system, yellow bar: metabolism; light blue bar: genetic information processing; orange bar: environmental information processing; light purple bar: cellular processes; light green bar: organismal systems; cyan blue: human diseases. Red and white denote high and low scores, respectively. For trend column, red and green dots indicate the highest and lowest z-scores, respectively. The number of fish is for microarray and NGS are listed in Appendix A, there are 14 samples for microarray, 16 samples for NGS.

**Figure 6 cancers-11-01899-f006:**
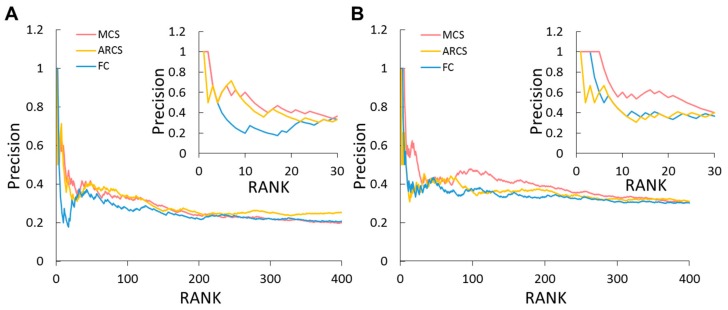
Precision comparison of three scoring methods for obesity and HCC. Scoring methods for selecting potential genes are maximum combined score (MCS, red), average root combined score (ARCS, orange), and fold change (FC, blue) for (**A**) obesity in WT fish and (**B**) HCC in HBx,Src(p53-) fish with overfeeding and high-fat diet. Positive sets are selected from DisGeNET database.

**Figure 7 cancers-11-01899-f007:**
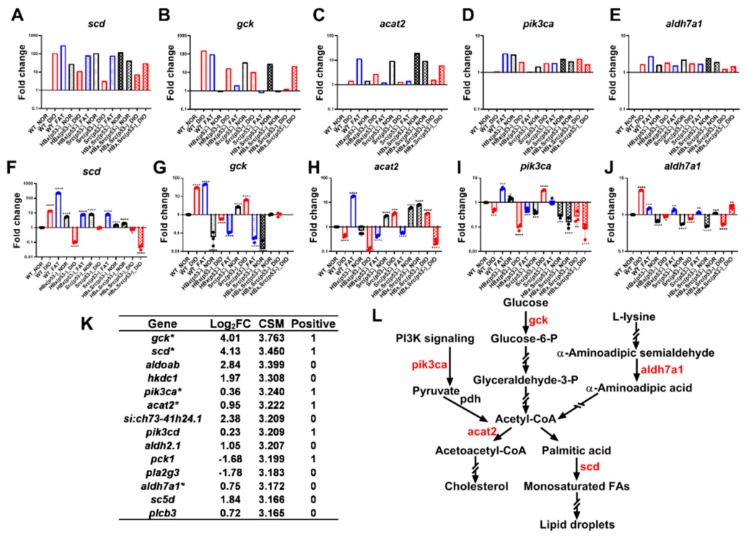
Validation of microarray with qPCR for the representative genes. Expression of (**A**,**F**) *scd*, (**B**,**G**) *gck*, (**C**,**H**) *acat2*, (**D**,**I**) *pik3ca*, and (**E**,**J**) *aldh7a1* in WT, HBx(p53-), Src(p53-), and HBx,Src(p53-) fish after eight weeks of normal diet (NOR), overfeeding (diet-induced obesity, DIO), or high-fat diet (FAT) from (**A**–**E**) microarray and (**F**–**J**) qPCR analysis. (**K**) Top 14 genes selected by global omics data analysis. (**L**) Among the 14 genes, 5 genes (red) overexpressed in WT diet-induced obesity are related to glucose and lipid metabolism. The number of fish is 20 for each group, and the number of experimental replicates for qPCR analysis is 3.

**Table 1 cancers-11-01899-t001:** Feeding Schedule.

	Group	Normal Diet (NOR)	Diet-Induced Obesity (DIO)	High-Fat Diet (FAT)
Time				
09:00		Powdered feed (0.22g)	powdered feed (0.22 g)	powdered feed (0.22 g)
11:00		None	brine shrimp 2 mL	high-fat fish food (0.22 g)
15:00		*Artemia* 0.5mL	brine shrimp 2 mL	high-fat fish food (0.22 g) brine shrimp 0.5 mL
17:30		None	brine shrimp 2 mL	high-fat fish food (0.22 g)

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
