# Peer review of "Omics-based Investigation of Diet-induced Obesity Synergized with HBx, Src, and p53 Mutation Accelerating Hepatocarcinogenesis in Zebrafish Model"

_cancers, 2019, doi:10.3390/cancers11121899_

Round 1

Reviewer 1 Report

Yuan WY, et al. reported Omics-based investigation of diet-induced obesity in zebrafish model. The study is interesting and well-performed , however, there are some concerns that should be addressed.

Major

The authors should describe the characteristics of HCCs. Were there any differences according to the diet?

The author should discuss the mechanism of NASH-HCC in human by using the results of this study and previous results in published manuscripts.

It is already well-known that HBx is associated with HCC, but in this study, with any diets, no HCC was detected in HBx (p53-) model. The author should discuss it.

The author should explain the reason why ccne1, cdk1, and cdk2 were used as cell cycle/proliferation markers.

Author Response

Reviewer #1:

Major comments:

Yuan WY, et al. reported Omics-based investigation of diet-induced obesity in zebrafish model. The study is interesting and well-performed.

Reply: Thank Reviewer #1 for acknowledging our works.

However, there are some concerns that should be addressed. The authors should describe the characteristics of HCCs. Were there any differences according to the diet?

Reply: Thanks to the reviewer for the valuable comments. We have added the description of the HCC in the Introduction section to the revised manuscript.

The major risk factors for HCC include hepatitis B and hepatitis C virus infection, aflatoxin contamination, as well as chronic alcohol consumption. HCC also has been associated with non-alcoholic fatty liver disease (NAFLD), nonalcoholic steatohepatitis (NASH), diabetes, and obesity. There is no differences on the characteristics of HCC according to the diet.

NAFLD and NASH are metabolic diseases which are major drivers of HCC, and diet induced obesity and high-fat diet is the cause of metabolic disorder. Obesity is closely related to diabetes, chronic liver disease, and many cancers. More importantly, obesity has also been identified as one of the main factors contributing to HCC (Gan et al., 2018; Ray, 2013; Sun and Karin, 2012). Metabolic risk factors such as fatty liver, high triglyceride levels, and diabetes mellitus are significantly associated with nonviral HCC in Taiwan (Huang et al., 2018).

Genetic variants associated with obesity can be modified by obesogenic environments such as increase of high fat, high sugar beverage consumption, and decrease of physical activity (Walter et al., 2016). However, the synergistic effects between diet-induced obesity and genetic risk factors for liver disease and liver cancer is unclear. It is essential to understand the synergism between obesity and genetic risk factors, and to develop therapeutic techniques derived from those discoveries. It has been demonstrated that zebrafish and mammalian diet-induced obesity are through a similar mechanism (Oka et al., 2010). We therefore used zebrafish model and omics study trying to solve this mystery.

We have compared all HCCs from the different genetic background and diets to see how the genetic influence the HCC biology in Figure 4. Actually, the extensive diet treatment for WT, HBx(p53-) and Src(p53-) promoted HCC formation, however, the HBx,Src(p53-) diet-induced obesity for 16 weeks reduced the HCC formation may due to the self-healing of zebrafish reported from various transgenic fish lines from our lab and others (Chou et al., 2019; Lu et al., 2013; Mudbhary et al., 2014; Su et al., 2019; Tu et al., 2017). When we compared the HCC biology from different models, we can see the HBx,Src(p53-) diet induced obesity for 8 weeks exhibit the most intensive HCC characteristics with more hepatocytes. 

The author should discuss the mechanism of NASH-HCC in human by using the results of this study and previous results in published manuscripts.

Reply: Thanks to the reviewer for the valuable comments. We have added the discussion as below to the discussion section in this revised manuscript.

The progression from NAFLD to NASH in human was original proposed as “two-hit hypothesis” (Day and James, 1998) in which insulin resistance mediated increase of free fatty acids due to enhanced lipolysis was the first hit that leads to steatosis. The increase level of fatty acid oxidation enhancing oxidative stress was the second hit that triggers lipid peroxidation, inflammation, fibrosis and carcinogenesis. Due to the “two-hit hypothesis”  is insufficient to explain the complicated mechanisms in NAFLD/NASH-HCC, the “multiparallel-hits hypothesis” was proposed (Tilg and Moschen, 2010), and has recognized as the mechanism of NASH-HCC in human (Takakura et al., 2019), in which hepatic inflammation was the first cause, and numerous conditions (including genetic variations, abnormal lipid metabolism, oxidative and/or endoplasmic reticulum stress (ER stress), mitochondrial dysfunction, altered immune responses, and imbalance in gut microbiota) acting in parallel.

In the molecular levels, complex changes in signaling pathways due to genetic, epigenetic mediates metabolism dysregulation and cell proliferation (Kutlu et al., 2018). The proinflammatory cytokines IL-6 activates IAK/STAT3, phosphatidylinositol 3-kinases (PI3K)/AKT/mTOR, mitogen-activate protein kinase (MAPK) pathway and TGF-β, Wnt/β-catenin that regulate proliferation and energy metabolism in the cell were reported.

The combination of endoplasmic reticulum stress and high fat diet (HFD) can lead to HCC through a number of underlying mechanisms. HFD can produce moderate ER stress, which increases lipogenesis and hepatic steatosis, while they increase reactive oxygen species (ROS) and oxidative stress and subsequent causing genomic instability, leading to death of hepatic cell and release of inflammatory factors that stimulate hepatocyte proliferation lead to HCC (Takakura et al., 2019).

Using mouse model fed with high-fat-non-cholesterol versus high-fat-high-cholesterol, scientist had found high-cholesterol promotes NASH development. Upregulation of the metabolic genes (ALDH18A1, CAD, CHKA, POLD4, PSPH, and SQLE) and aberrant expression of cancer-related genes (ALCAM, ITGA6, DDIT3, MAP3K6 and PAK1) were found in mouse fed with high-fat-high-cholesterol similar to human NASH-HCCs (Liang et al., 2018). Another female mice model fed with western diet (WD) induced NASH increased the expression of genes including steatosis (SFA, MUFA, MUFA-containing di- and triacylglycerols), inflammation (TNFα), oxidative stress (Ncf2), and fibrosis (Col1A) via lipidomic and transcriptomic approach (Garcia-Jaramillo et al., 2019). Our results showed that pathways of fatty acid metabolism and steroid biosynthesis are activated during hepatocarcinogenesis which is consistent with previous results. In our results, we also found glycolysis/gluconeogenesis, insulin signaling pathway, and insulin resistance pathways were increased in WT and HBx(p53-) DIO fish. Our results also identified ribosome biogenesis are activated during hepatocarcinogenesis which was not mentioned from previous studies. Moreover, we identified genes related to glucose and lipid metabolism are overexpressed in WT diet-induced obesity, reveal glucose uptake from overfeeding will link to lipogenesis, this finding is novel and might explain the underlines mechanisms for overfeeding causing NASH.

It is already well-known that HBx is associated with HCC, but in this study, with any diets, no HCC was detected in HBx (p53-) model. The author should discuss it.

Reply: Thanks to the reviewer for the comment, we have added the discussion (as below) to the result section (subsection 2.6. Liver pathology after diet-induced obesity).

From our previous study, HBx(p53-) transgenic fish developed HCC at 11 months under normal diet (Lu et al., 2013). In this study, we fed the zebrafish for different diets at 3 months old for two months, and sacrificed at the age of 5 months. We observed about 40% hyperplasia and no HCC, which is similar to what was found previously. Previously we found Src(p53-) transgenic fish developed HCC from 7 to 11 months, and only have hyperplasia at 5 months (Lu et al., 2013). In this study, we also found Src(p53-) transgenic fish developed hyperplasia at 5 month of age which is similar to previous study.

In this revised manuscript, we have added the histopathological results from extended feeding in Figure 4. We fed the zebrafish for different diets at 3 months for four months, and scarified at the age of 7 months. With diet-induced obesity of high fat diet, HCC was detected in HBx(p53-), but no HCC was detected in the normal diet. However, HCC was detected in Src(p53-) fish from normal diet, diet-induced obesity or high-fat diet, with higher proportional fish developed HCC in DIO or HFD.

The author should explain the reason why ccne1, cdk1, and cdk2 were used as cell cycle/proliferation markers.

Reply: Thanks to the reviewer for the comment. Unlimited replicative potential is one of the hallmarks of cancer cells. Because cancer cells have the properties of uncontrolled cell proliferation and dysfunction of cell cycle checkpoint, we analyzed the cell cycle-related genes/proliferation marker genes G1/S-specific cyclin-E1 (ccne1), cyclin-dependent kinase 1 (cdk1), and cyclin-dependent kinase 2 (cdk2) by qPCR.

Reviewer 2 Report

The authors have to be congratulated to present an original study related to interactions between diet, obesity, genetic, and hepatocarcinogenesis. This is a highly relevant topic in human that lacks animal models to better understand the biological processes involved. Thus, such a study is very well timed. The authors use here various diets; various genetically modify zebrafish, and various technics to tackle nicely and comprehensively this problematic. The manuscript is clear, the study is well designed, the results are convincing. The representations of data are also of good quality and the supplementary figures are useful and well distributed. I have only minor comments.

It would be nice to indicate the number of replicate/experiments (n=?) in the figure legend.

If possible, it would be very informative to compare in a figure all HCC from the different models to see how the genetic influence the HCC biology. (gene expression histology …)

Figure 4 a, indication of standard deviation / spread could be informative if possible

In supplementary data, immune-deconvolution of Microarray and RNA-seq could be interesting to understand how the immune system is affected by different parameters.

Author Response

Reviewer #2:

Major comments:

The authors have to be congratulated to present an original study related to interactions between diet, obesity, genetic, and hepatocarcinogenesis. This is a highly relevant topic in human that lacks animal models to better understand the biological processes involved. Thus, such a study is very well timed. The authors use here various diets; various genetically modify zebrafish, and various technics to tackle nicely and comprehensively this problematic. The manuscript is clear, the study is well designed, the results are convincing. The representations of data are also of good quality and the supplementary figures are useful and well distributed.

Reply: Thank you for acknowledging our works.

I have only minor comments. It would be nice to indicate the number of replicate/experiments (n=?) in the figure legend.

Reply: Thanks to the reviewer for the constructive comments. In the Materials and Methods section 4.1. Zebrafish Maintenance and Transgenic Zebrafish Lines, we have mentioned “four types of fish, 3 months old, were treated with 3 feeding methods: normal diet, overfeeding, or high-fat diet. After 8 weeks of feeding, fish were weighed, and liver tissue from about 20 fish was collected.” So, the number of fish is 20, and the number of experimental replicates for qPCR analysis is 3. We will add those information in the legends of Figures 2, 3, 5, 7, S1, S2, S3, S4.

If possible, it would be very informative to compare in a figure all HCC from the different models to see how the genetic influence the HCC biology. (gene expression histology …)

Reply: Thanks to the reviewer for the constructive comments. We have compared all HCCs from the different genetic background and diets to see how the genetic influence the HCC biology in Figure 4. Actually, the extensive diet treatment for WT, HBx(p53-) and Src(p53-) promoted HCC formation, however, the HBx,Src(p53-) diet-induced obesity for 16 weeks reduced the HCC formation may due to the self-healing of zebrafish reported from various transgenic fish lines from our lab and others (Chou et al., 2019; Lu et al., 2013; Mudbhary et al., 2014; Su et al., 2019; Tu et al., 2017). When we compared the HCC biology from different models, we can see the HBx,Src(p53-) diet induced obesity for 8 weeks exhibit the most intensive HCC characteristics with more hepatocytes.

Figure 4a, indication of standard deviation / spread could be informative if possible.

Reply: Thanks to the reviewer for the valuable comments. We have redrawn the Fig. 4a and added the standard deviation in each condition for analyzing the data consistency in system level. In this revised manuscript, we add a new figure as figure 4. Therefore, the original figure 4 becomes figure 5 in this revised manuscript.

We calculated the meta-z-scores for each system derived from their z-scores of involving pathways to evaluate their significance. Therefore, we computed the standard deviation using z-scores of involving pathways in six systems with four genetic backgrounds and two types data (i.e., microarray and RNA-seq). We found Src(p53-) and HBx,Src(p53-) in genetic information processing have the bigger standard deviation than other conditions in microarray and RNA-seq. The results may cause by the number of involving pathways and distance of z-scores. For example, the system of genetic information processing has only 22 involving pathways (metabolism: 184 involving pathways), but the z-scores of involving pathways have big distance contributed by the pathway of ribosome biogenesis in eukaryotes (e.g., Max: 7.638, Ave: 1.00 in HBx,Src(p53-)). Based on the equation of standard deviation, less involving pathways may get the bigger standard deviation.

In supplementary data, immune-deconvolution of Microarray and RNA-seq could be interesting to understand how the immune system is affected by different parameters.

Reply: Thank you for your constructive comments. In this revised manuscript, we have added the results of immune related systems affected by four genetic background zebrafish based on KEGG database to the discussion section.

Based on 26 significantly consistent pathways (p-value < 0.05) in both MIC and NGS (Fig. 5C), some of these pathways are related to immune responses in four genetic background zebrafish and we summarized the observations as follows: First, toll-like receptor signaling pathway, playing key roles in the immune system, consistently decreased in WT, HBx(p53-), Src(p53-), and HBx,Src(p53-) fish. We found the gene expressions of tlr5a and tlr5b were up-regulated in WT, and gene nfkbiaa was up-regulated in both WT and HBx(p53-). These genes are highly related with immune response. In addition, several pathways (e.g., NOD-like receptor signaling pathway and RIG-I-like receptor signaling pathway), which belong to immune system based on KEGG database, have the similar trends in omics data. Second, the insulin signaling and insulin resistance pathways were considered to participate in the regulation of islet endocrine influenced by immune system (Dalmas, 2019; Zhang et al., 2019). They increased in WT and HBx(p53-) fish and decreased in Src(p53-) and HBx,Src(p53-) fish. Third, the pathway of proteasome consistently decreased in WT and HBx(p53-) fish and increased in Src(p53-) and HBx,Src(p53-) fish. It regulates the immune system by degrading immune and inflammatory mediators (Kammerl and Meiners, 2016). Interestingly, the trend of the proteasome is opposite to the trend of the toll-like receptor signaling pathway. Final, the metabolic pathways (e.g., fatty acid degradation) consistently increased from WT, HBx(p53-), Src(p53-) to HBx,Src(p53-) fish. Immune responses could be potentially modified by fatty acids, and the modifications include the organization of lipids in the cells and interaction with nuclear receptors (Yaqoob, 2004). These results imply that immune system plays the key role for diet-induced obesity and accelerating hepatocarcinogenesis in zebrafish.

Reviewer 3 Report

In this paper, the synergism between diet and genetic risk factors in hepatocarcinogenesis, were investigated, using zebrafish with four genetic backgrounds and obesity induced by overfeeding or high-fat-diet-induced,  with omics-based expression of genes and histopathological changes. It was found that  overfeeding and high-fat diet induce obesity and nonalcoholic steatohepatitis in wild-type fish. In HBx,Src(p53-) triple transgenic zebrafish, the diet-induced obesity enhanced the development of HCC and increased the cancer incidence. The authors developed a global omics data analysis method to investigate genes, pathways, and biology systems based on microarray and next- generation sequencing (NGS, RNA-seq) omics data of zebrafish with four diet and genetic risk factors. It was found the activation, during hepatocarcinogenesis, of metabolism and genetic information processing, and pathways of fatty acid metabolism, steroid biosynthesis, and ribosome biogenesis.

The major weakness of this paper, is that potential genetic changes were investigated by omic technology without adequate experimental control. Validation experiments, largely insufficient, were only made by mRNA analysis. The relationships between the genes overexpressed and glucose and lipid metabolism must be also evaluated at protein and activity levels. In the absence of these evaluations the conclusions are quite speculative.

Some minor points: some particulars of the Fig.s 2, 3 and 6 are too small and difficult to read.

Author Response

Reviewer #3:

Major comments:

In this paper, the synergism between diet and genetic risk factors in hepatocarcinogenesis, were investigated, using zebrafish with four genetic backgrounds and obesity induced by overfeeding or high-fat-diet-induced, with omics-based expression of genes and histopathological changes. It was found that overfeeding and high-fat diet induce obesity and nonalcoholic steatohepatitis in wild-type fish. In HBx,Src(p53-) triple transgenic zebrafish, the diet-induced obesity enhanced the development of HCC and increased the cancer incidence. The authors developed a global omics data analysis method to investigate genes, pathways, and biology systems based on microarray and next- generation sequencing (NGS, RNA-seq) omics data of zebrafish with four diet and genetic risk factors. It was found the activation, during hepatocarcinogenesis, of metabolism and genetic information processing, and pathways of fatty acid metabolism, steroid biosynthesis, and ribosome biogenesis.

Reply: Thank you for your comments.

The major weakness of this paper, is that potential genetic changes were investigated by omic technology without adequate experimental control. Validation experiments, largely insufficient, were only made by mRNA analysis. The relationships between the genes overexpressed and glucose and lipid metabolism must be also evaluated at protein and activity levels. In the absence of these evaluations the conclusions are quite speculative.

Reply: Thank you for your comments. Using microarray and RNA-seq, we analyzed the transcriptome changes for different genetic background fed with various diets, and the control for different genetic background and diet treatment was the WT fish with normal diet. Since our data is RNA changes from microarray or RNA-seq, we verify the results using qPCR analysis for more fish with the same genetic background and diet treatment. The comprehensive and systematic evaluations required proteomics, metabolomics and systematic analysis which is the goal of our current research.

Some minor points: some particulars of the Fig.s 2, 3 and 6 are too small and difficult to read.

Reply: Thank you for your comments, we have made the words bigger and easier to read.

Reviewer 4 Report

This paper aims to present a general view of the interaction and possible synergism between genetic and diet factors during the induction of liver cancer process. The study has been developed employing an animal (Zebra fish) model of disease, using three types of diets and four types of genetic backgrounds. Authors have employed several multidisciplinary techniques, combined with microarrays and next-generation sequencing. Although some results obtained could be expected to some point, it is always important to confirm them. Thus, taking into account that they have employed three types of different diets (normal, DIO; diet-induced obesity and FAT; high fat diet) and genetic backgrounds related to hepatocellular carcinoma, it is reasonable that they have mainly observed differences in genes related to “metabolism” and “genetic information”.

The novelty of the work resides in that authors describe the relation between diet and genetic factors employing omics technology, and that they propose a new system for the omics data analysis.

General comments:

Some of the paragraphs of this version are marked in yellow. Perhaps it is not the final version of the manuscript. If authors could present the incidence in humans of the different genetic backgrounds employed in the study HBx(p53-), Src(p53-) and HBx,Src(p53-) triple transgenic, they could better reflect the impact of the study. Section 2.1 of the Results. The entire section (including figure 1) explains the procedure and experimental design of the study. No result is present in this section. Even though it results clarifying, authors should consider to include this section in the supplementary data section.   Some paragraphs in the “Results” section contains, compare and contrast results from this study with other papers from the same group or from others groups, or in some other cases authors speculate about the results obtained. Authors should consider including those paragraphs in the Discussion section. i.e: paragraph from lines 210 to 215; paragraph from lines 216 to 223; lines 298 to 300; lines 309 to 310. The conclusion presented for Section 2.4 of the results (“Overall, DIO or FAT increased the expression of lipogenic factors and lipogenic enzymes, and the expression of cell cycle related genes was hardly increased after eight weeks of DIO or FAT.”) is too general and does not properly reflect the results obtained. In this sense, some of the results presented are contradictory with this general statement. As an example, in line 159 it says that “DIO did not cause differences in the expression of agpat and pap, but decreased the expression of fasn.” Thus, two lipogenic enzymes (out of three) did not change and the other lipogenic enzyme was in fact decreased due to the DIO.

Regarding the lipogenic factors, in line 163 it says that “The expression of chrebp was decreased in both FAT and DIO after eight 163 weeks”. The authors analyse three factors and one of them are in fact decreased due to the FAT and DIO diets.

Some results that reflects the significant differences in males/females are presented in Supplementary data. Authors could consider to include some of those results as figures of the article. The statement of line 184 “there was no significant difference in the expression of cell cycle–related genes compared to normal feeding” does not correspond to what can be observed in Figure 2F. Apparently, according to the figure, FAT diet significantly decreased the values of ccne1 and cdk1. All experiments of Figure 2 have been developed with NOR, DIO and FAT diet except those performed with the triple transgenic fish, from figures 2 J to L (that includes only the NOR and DIO groups). Authors should justify/mention the reason why they have omitted the FAT group. Regarding Figure 3, the text in line 205 states that “DIO and FAT caused slight hyperplasia”, and in Figure 3 B it reflects that WT fish fed with DIO presents “hyplerplasia”. However, in Figure 3 D, it reflects that at 5 months, 95 % of the livers were “Normal”. Authors should explain the differences observed. All experiments of Figure 3 have been developed with NOR, DIO and FAT diet except those performed with the triple transgenic fish, from figures 3 M to O (that includes only the NOR and DIO groups). Authors should justify/mention the reason why they have omitted the FAT group. All experiments of Figure 4 have been developed with NOR, DIO and FAT diet except those performed with the triple transgenic fish, from figures 4 D (that includes only the NOR and DIO groups). Authors should justify/mention the reason why they have omitted the FAT group. Figure 4A, represents a “Representative images of HCCs from various groups”. It is not clear the criteria employed by the authors in the selection of the images/groups.

Thus, in the first line they present four images from the triple transgenic fish, one fed with NOR diet and three with DIO diet.

In the second line they represent all the different genotypes, but fed with different diets (three DIO and one NOR)

In the third line they represent all the different genotypes, but fed with different diets (three FAT and one NOR).

Table S1 mentioned in line 241 of the text, presents the Experimental design of microarray and next-generation sequencing. Apparently some the groups include 1 sample and other groups, 2 samples. Authors should explain the criteria employed in the selection of number of samples/per group. Figure 5. This referee and perhaps other potential readers would appreciate if authors could explain what group of genes they include in the “System” called “Human Diseases”, analysed in Figure 5. Only one gene is mentioned in this particular System. It is surprising not to find more genes as they induce in some cases HCC. Page 11, Line 337. Authors should clarify the state: “The results show that MCS and ARCS performed better than fold change for obesity and HCC” It is not clear for this referee the contribution to the work of Figure 6 (Precision comparison vr rank of three scoring methods). It is not clear what is represented in the figure. Page 13 line 381. Authors mentioned that “hyperplasia was already obvious with a normal diet and did not increase by diet”. According to figure 3, a DIO diet can increase Page 15 “Feeding method”. The text results confusing. A table including the precise times and quantities and that could be included in the Supplementary data could help readers to understand the feeding method. Page 17. Statistical methods. The works does not include some interaction studies between genotypes and diets.

Minor comments.

Page 2 Line 86. Paragraph 4 of this page ends describing the present work. However, the next paragraph starts with “most of these studies…”. It is confusing. Page 3 Line 112. There is a gap after cdk2. Page 5 Line162. The expression of srebf1 was more significant in FAT than DIO”. It should state if there is a significant increase or a decrease. Page 11 Line 321. There are two “and” together. Figure 7. It is difficult to appreciate and read the treatments represented if figure 7 A to J. Moreover, apparently there are two treatments (HBx, Src(p53) NOR and HBx, Src(p53) DIO) that they are repeated twice. Figure 7. The footnote of the figure indicates that there are “12 Top genes”. In Figure 7 table K, there are 14 genes. It also happens in line 346.

Page 13 Line 381. It should be “ steatosis measured by H&E”. Page 16 lines 485-486. “All of the fish were maintained in the Zebrafish Core Facility at the National Health Research Institute (NHRI), Taiwan”. It is already mentioned in Page 15 line 460 Page 16 lines 486-487. “All experiments involving zebrafish were approved by the Institution Animal Care and Use Committee (IACUC) of the NHRI.” It is already mentioned in Page 15 line 462. Page 19 lines 602; In the text ….“is defined as”, something is missing in this line. Bibliography: The DOI number of references 18, 24 and 63 are missing.

Author Response

Reviewer #4

This paper aims to present a general view of the interaction and possible synergism between genetic and diet factors during the induction of liver cancer process. The study has been developed employing an animal (Zebra fish) model of disease, using three types of diets and four types of genetic backgrounds. Authors have employed several multidisciplinary techniques, combined with microarrays and next-generation sequencing. Although some results obtained could be expected to some point, it is always important to confirm them. Thus, taking into account that they have employed three types of different diets (normal, DIO; diet-induced obesity and FAT; high fat diet) and genetic backgrounds related to hepatocellular carcinoma, it is reasonable that they have mainly observed differences in genes related to “metabolism” and “genetic information”. The novelty of the work resides in that authors describe the relation between diet and genetic factors employing omics technology, and that they propose a new system for the omics data analysis.

Reply: Thank you for acknowledging our works.

Some of the paragraphs of this version are marked in yellow. Perhaps it is not the final version of the manuscript.

Reply: Thank you very much for your comments. Actually, the yellow-highlighted are the revised parts which were not in our final version, probably marked by the editorial office to let the reviewers see clearly.

If authors could present the incidence in humans of the different genetic backgrounds employed in the study HBx(p53-), Src(p53-) and HBx,Src(p53-) triple transgenic, they could better reflect the impact of the study.

Reply: Thank you very much for the advice and suggestions. We have added those information in the introduction section: In human HCC patients, 75% of HCC cancer tissue were HBx positive [1], and 65.38% of HCC tissue were SRC positive in Chinese population [2]. HBx and TP53 R249S mutation were found in 77% HCC patients in West African population [3]. Therefore, we generated transgenic zebrafish model to reflect the genetic signature in human HCC patients.

Section 2.1 of the Results. The entire section (including figure 1) explains the procedure and experimental design of the study. No result is present in this section. Even though it results clarifying, authors should consider to include this section in the supplementary data section.  

Reply: Thank you for your valuable comments. In this revised manuscript, we rewrote the paragraph to improve the readability. Basically, we added summarized results and observations in this section. For example, we added the following statement: “When compared to WT with normal diet, DIO or FAT increased the expression of lipogenic factors and lipogenic enzymes, including 1-acylglycerol-3-phosphate acyltransferase (agpat), fatty acid synthase (fasn) and phosphatidate phosphatase (pap) by using qPCR (Fig. 1C) and hematoxylin and eosin (H&E) (Fig. 1D).” in this revised manuscript.

Some paragraphs in the “Results” section contains, compare and contrast results from this study with other papers from the same group or from others groups, or in some other cases authors speculate about the results obtained. Authors should consider including those paragraphs in the Discussion section. i.e: paragraph from lines 210 to 215; paragraph from lines 216 to 223; lines 298 to 300; lines 309 to 310

Reply: Thank you very much for the advice and suggestions. Those paragraphs were meant to answering reviewers’ question, and make it clear. We have move the paragraph from lines 210 to 215; paragraph from lines 216 to 223; lines 298 to 300; lines 309 to 310 to the Discussion section.

The conclusion presented for Section 2.4 of the results (“Overall, DIO or FAT increased the expression of lipogenic factors and lipogenic enzymes, and the expression of cell cycle related genes was hardly increased after eight weeks of DIO or FAT.”) is too general and does not properly reflect the results obtained. In this sense, some of the results presented are contradictory with this general statement. As an example, in line 159 it says that “DIO did not cause differences in the expression of agpat and pap, but decreased the expression of fasn.” Thus, two lipogenic enzymes (out of three) did not change and the other lipogenic enzyme was in fact decreased due to the DIO. Regarding the lipogenic factors, in line 163 it says that “The expression of chrebp was decreased in both FAT and DIO after eight 163 weeks”. The authors analyse three factors and one of them are in fact decreased due to the FAT and DIO diets.

Reply: Thank you for the suggestion. We have remove the general conclusion at the end.

Some results that reflects the significant differences in males/females are presented in Supplementary data. Authors could consider to include some of those results as figures of the article.

Reply: The gender differences is not the main focus of this article, and if we include those results in the main article, there will be too many figures, so we presented those in the supplementary figures.

The statement of line 184 “there was no significant difference in the expression of cell cycle–related genes compared to normal feeding” does not correspond to what can be observed in Figure 2F. Apparently, according to the figure, FAT diet significantly decreased the values of ccne1 and cdk1.

Reply: Thank you for the advice. We have edited this statement to correspond to the observation.

All experiments of Figure 2 have been developed with NOR, DIO and FAT diet except those performed with the triple transgenic fish, from figures 2 J to L (that includes only the NOR and DIO groups). Authors should justify/mention the reason why they have omitted the FAT group.

Reply: Thank you for the comments. We have justified the reason for using only DIO for treated the triple transgenic fish. From previous experiments, we found DIO and FAT diet exhibited similar effect on zebrafish, so for the triple transgenic fish, we only applied DIO diet to the fish.

Regarding Figure 3, the text in line 205 states that “DIO and FAT caused slight hyperplasia”, and in Figure 3 B it reflects that WT fish fed with DIO presents “hyperplasia”. However, in Figure 3D, it reflects that at 5 months, 95 % of the livers were “Normal”. Authors should explain the differences observed.

Reply: Thank you for the comments. We observed one fish out of 20 fish in WT developed hyperplasia after FAT and DIO, we have re-written the statement more accurately, and Figure 3B is the representative image. In WT fish, FAT increased steatosis, and both DIO and FAT caused slight hyperplasia (one fish out of 20 fish developed hyperplasia), which was more significant with prolonged feeding (Fig. 4D).

All experiments of Figure 3 have been developed with NOR, DIO and FAT diet except those performed with the triple transgenic fish, from figures 3 M to O (that includes only the NOR and DIO groups). Authors should justify/mention the reason why they have omitted the FAT group.

Reply: Thank you for the comments. We have justified the reason for using only DIO for treated the triple transgenic fish. From previous experiments, we found DIO and FAT diet exhibited similar effect on zebrafish, so for the triple transgenic fish, we only applied DIO diet and normal diet to the fish.

All experiments of Figure 4 have been developed with NOR, DIO and FAT diet except those performed with the triple transgenic fish, from figures 4 D (that includes only the NOR and DIO groups). Authors should justify/mention the reason why they have omitted the FAT group.

Reply: Thank you for the comments. We have justified the reason for using only DIO for treated the triple transgenic fish. From previous experiments, we found DIO and FAT diet exhibited similar effect on zebrafish, so for the triple transgenic fish, we only applied DIO diet and normal diet to the fish.

Figure 4A, represents a “Representative images of HCCs from various groups”. It is not clear the criteria employed by the authors in the selection of the images/groups. Thus, in the first line they present four images from the triple transgenic fish, one fed with NOR diet and three with DIO diet. In the second line they represent all the different genotypes, but fed with different diets (three DIO and one NOR) In the third line they represent all the different genotypes, but fed with different diets (three FAT and one NOR).

Reply: Thank you very much for the advice and suggestions. This figure was added during revision, to answer the reviewers’ request to compare the HCC from different genetic background and diet. We have re-made the Figure 4 as the format of Figure 3, to use representative images for different diet treatment.

Table S1 mentioned in line 241 of the text, presents the Experimental design of microarray and next-generation sequencing. Apparently some the groups include 1 sample and other groups, 2 samples. Authors should explain the criteria employed in the selection of number of samples/per group.

Reply: Thank you very much for the comments. We selected the fish specimens based on the histopathological features and qPCR pattern, we included two fish for normal diet in WT, one fish for HBx,(p53-) –NOR and Src(p53-)-NOR because they have similar histopathological features and qPCR pattern, so we could group them together.

Figure 5. This referee and perhaps other potential readers would appreciate if authors could explain what group of genes they include in the “System” called “Human Diseases”, analysed in Figure 5. Only one gene is mentioned in this particular System. It is surprising not to find more genes as they induce in some cases HCC.

Reply: Thank you for your constructive comments. In this revised manuscript, we added two new Tables (Tables S2 and S3) to illustrate the involved DEGs in insulin resistance pathway of human diseases in the four types of fish between normal and obesity diet (Fig. 5). For microarray data, the number of involved DEGs in this pathway are 11 (WT), 24 (HBx(p53-)), 16 (Src(p53-)), and 13 (HBx,Src(p53-)). Additionally, the number of involved DEGs in NGS are 6 (WT), 8 (HBx(p53-)), 10 (Src(p53-)), and 13 (HBx,Src(p53-)). For examples, several DEGs selected from microarray were related to HCC, such as ppargc1a [4] and trib3 [5].

Page 11, Line 337. Authors should clarify the state: “The results show that MCS and ARCS performed better than fold change for obesity and HCC” It is not clear for this referee the contribution to the work of Figure 6 (Precision comparison vr rank of three scoring methods). It is not clear what is represented in the figure.

Reply: Thank you for your valuable comments. In this revised manuscript, we added two new tables (Tables S4 and S5) to show top ranked 20 genes of MCS and the corresponding ranks of ARCS and FC in obesity and HCC, respectively. Among these 20 genes, 8 genes (e.g., gck, scd, and pik3ca) are related to obesity recorded in DisGeNET (Table S4). Conversely, 7 and 5 genes are recorded in DisGeNET for top ranked 20 genes from ARCS and FC, respectively. For top ranked 20 genes of MCS, 11 genes (e.g., sqlea, hmgcra, and mvda) are related to HCC (Table S5). In contrast to the MCS, 7 out of top ranked 20 genes from both ARCS and FC are recorded in DisGeNET.

Page 13 line 381. Authors mentioned that “hyperplasia was already obvious with a normal diet and did not increase by diet”. According to figure 3, a DIO diet can increase

Reply: Thank you for the comments, we have modified this statement as DIO diet can increase the hyperplasia a little bit in HBx(p53-) fish.

Page 15 “Feeding method”. The text results confusing. A table including the precise times and quantities and that could be included in the Supplementary data could help readers to understand the feeding method.

Reply: Thank you for the suggestion, we have re-written the feeding method and including the table with precise time and quantities for the feeding method.

Page 17. Statistical methods. The works does not include some interaction studies between genotypes and diets.

Reply: Thank you for the comments, this work analyzed the body weight and qPCR data using SPSS 17.0 (SPSS, Inc.) and Prims GraphPad for the statistical significance for the different genetic background under various diet. We normalized with WT(NOR).

Minor comments:

Page 2 Line 86. Paragraph 4 of this page ends describing the present work. However, the next paragraph starts with “most of these studies…”. It is confusing.

Reply: We reorganized the paragraph and rewrote some statements to improve the readability.

Page 3 Line 112. There is a gap after cdk2. Page 5 Line162. The expression of srebf1 was more significant in FAT than DIO”. It should state if there is a significant increase or a decrease.

Reply: Thanks for your careful checking, we have corrected this mistake.

Page 11 Line 321. There are two “and” together.

Reply: We added the missing word, SiSj and SiSj, in the manuscript and removed the first “and”.

Figure 7. It is difficult to appreciate and read the treatments represented if figure 7 A to J. Moreover, apparently there are two treatments (HBx, Src(p53) NOR and HBx, Src(p53) DIO) that they are repeated twice.

Reply: Thank you for the comments. Figure 7 is for comparison the microarray data (Fig. 7A-E) versus qPCR (Fig. 7F-J), the microarray contains only selected samples, we validated the mRNA expression levels using qPCR, so we make sure the expression patterns reflect the effect of diet/genotypes. For the triple transgenic fish, we found the two NOR diet and two DIO fish has different expression pattern, so we displayed them separately.

Figure 7. The footnote of the figure indicates that there are “12 Top genes”. In Figure 7 table K, there are 14 genes. It also happens in line 346.

Reply: Thanks for your careful checking, we have corrected this mistake.

Page 13 Line 381. It should be “steatosis measured by H&E”.

Reply: Thanks for your careful checking, we have corrected this mistake.

Page 16 lines 485-486. “All of the fish were maintained in the Zebrafish Core Facility at the National Health Research Institute (NHRI), Taiwan”. It is already mentioned in Page 15 line 460 Page 16 lines 486-487. “All experiments involving zebrafish were approved by the Institution Animal Care and Use Committee (IACUC) of the NHRI.” It is already mentioned in Page 15 line 462.

Reply: We removed the two repeated sentence in section 4.2.

Page 19 lines 602; In the text ….“is defined as”, something is missing in this line.

Reply: We added the missing word, SFC, in the manuscript.

Bibliography: The DOI number of references 18, 24 and 63 are missing.

Reply: We checked several papers published by cancers journal and confirmed the MDPI format is not including the DOI number, so we reapplied the MDPI format to all references.

References:

Shi, D.R. [The HBx protein expression in liver cancer]. Zhonghua Bing Li Xue Za Zhi 1991, 20, 85-87. Zhao, R.; Wu, Y.; Wang, T.; Zhang, Y.; Kong, D.; Zhang, L.; Li, X.; Wang, G.; Jin, Y.; Jin, X., et al. Elevated Src expression associated with hepatocellular carcinoma metastasis in northern Chinese patients. Oncol Lett 2015, 10, 3026-3034. Gouas, D.A.; Villar, S.; Ortiz-Cuaran, S.; Legros, P.; Ferro, G.; Kirk, G.D.; Lesi, O.A.; Mendy, M.; Bah, E.; Friesen, M.D., et al. TP53 R249S mutation, genetic variations in HBX and risk of hepatocellular carcinoma in The Gambia. Carcinogenesis 2012, 33, 1219-1224. Zhang, S.; Jiang, J.; Chen, Z.; Wang, Y.; Tang, W.; Chen, Y.; Liu, L. Relationship of PPARG, PPARGC1A, and PPARGC1B polymorphisms with susceptibility to hepatocellular carcinoma in an eastern Chinese Han population. OncoTargets and therapy 2018, 11, 4651-4660. Vara, D.; Morell, C.; Rodriguez-Henche, N.; Diaz-Laviada, I. Involvement of PPARgamma in the antitumoral action of cannabinoids on hepatocellular carcinoma. Cell death & disease 2013, 4, e618.

Round 2

Reviewer 1 Report

The author sincerely responded to the reviewers' comments and this manuscript is worth publishing. 

Author Response

Review #1

The author sincerely responded to the reviewers' comments and this manuscript is worth publishing.

Reply: Thank you very much for acknowledging our works.

Reviewer 3 Report

Although the authors introduced numerous changes in their paper, the latter still presents numerous drawbacks.

What they call “statistical analysis” in the legends of Figures 3 and 4 is merely a calculus of the percentages of histologic changes (HCC, hyperplasia, dysplasia, etc.). How many samples have been examined and the differences between the mean values  are statistically significant? The molecular analysis, merely based on mRNA expression, showed differences in expression of five enzymes involved in Pi3K signaling, glucose and palmitic acid metabolism. These findings do not support the conclusions of the authors relative to the role of these changes in obesity and HCC (lines 358-31). At least the expression of the proteins involved must be determined and the metabolic cycles should be evaluated. The Discussion section needs accurate revision. For instance, the sentences “hyperplasia was already obvious with a normal diet and did not increase by diet”, “….inflammation was the  first cause, and numerous conditions (including genetic variations, abnormal lipid metabolism, oxidative and/or endoplasmic reticulum stress (ER stress), mitochondrial dysfunction, altered  immune responses, and imbalance in gut microbiota) acting in parallel”, etc. are verbose and   The English form also needs accurate revision.

Author Response

ad replied to the comments.

Review #3

Although the authors introduced numerous changes in their paper, the latter still presents numerous drawbacks. What they call “statistical analysis” in the legends of Figures 3 and 4 is merely a calculus of the percentages of histologic changes (HCC, hyperplasia, dysplasia, etc.). How many samples have been examined and the differences between the mean values are statistically significant?

Reply: Thank you very much for the advice and suggestions. For Figure 3 and Figure 4, we should use “histopathological changes” instead of “statistical analysis”. Our focus was diet-induced obesity for two months, so the fish was 5 months old for the histopathological analysis (Fig. 3), there were 20 for WT, HBx(p53-), src(p53-) group, and 15 for the HBx,src(p53-). For the 7 months old (Fig. 4), we had only analyzed 4 to 7 fish for each group, and 15 for HBx,src(p53-) for comparison the effect of longer diet-induced obesity for last revision, we have re-analyzed all 20 fish for WT, HBx(p53-) and Src(p53-). Due to some specimens were missing at the Pathology core, the final fish number may not be exactly 20 or 15, the fish number are added to the figure legend.

The molecular analysis, merely based on mRNA expression, showed differences in expression of five enzymes involved in Pi3K signaling, glucose and palmitic acid metabolism. These findings do not support the conclusions of the authors relative to the role of these changes in obesity and HCC (lines 358-31). At least the expression of the proteins involved must be determined and the metabolic cycles should be evaluated.

Reply: Thank you very much for the comments. We have re-written this part to reflect our finding. Because antibody against zebrafish are rare, we could not find antibody against scd, gck, acat2, pik3ca and aldh7a1, so the proteins expression levels were not determined. And, because the fish already been sacrificed long time ago, so the metabolic cycles could not be evaluated. Due to the time limit, it is also not possible to raise the fish again and repeat the entire experiment. Please understand our current situation, and we just want to conclude that the upregulation of mRNA expression for the critical genes identified by microarray correlated to the steatosis during early hepatocarcinogenesis revealed by the histopathological diagnosis. Our global omics data analysis identified genes are connected to glucose and lipid metabolism also participating in NASH from the WT fish diet-induced obesity model (Fig. 7L), and those genes might be potential drug targets for prevention NASH, obesity and HCC.

The Discussion section needs accurate revision. For instance, the sentences “hyperplasia was already obvious with a normal diet and did not increase by diet”, “….inflammation was the first cause, and numerous conditions (including genetic variations, abnormal lipid metabolism, oxidative and/or endoplasmic reticulum stress (ER stress), mitochondrial dysfunction, altered immune responses, and imbalance in gut microbiota) acting in parallel”, etc. are verbose and the English form also needs accurate revision.

Reply: Thank you very much for the advice, for the Discussion section, we have accurate revised the Discussion section, and sent the revised manuscript for editing by MDPI English editing service.

Reviewer 4 Report

I would like to thank the authors for effort they have made to improve the manuscript. Authors have taken into consideration most of the suggestions made by the reviewer.

They have corrected those parts that were pointed and answered or discuss the questions proposed. Regarding the questions proposed by this referee, authors have correctly answered. They have also changed some figures and parts of the text that did not signficantly contribute for the potential readers of the work. Figure 1 still presents the procedure and experimental design of the study instead of results, but authors have the right to present their work in their way.

I think that some of the conclusions reflected in this new versión are more accurate or precise than the ones in the first manuscript.

Round 3

Reviewer 3 Report

The authors have satisfactorily met the reviewer suggestions.